# Defects in the cell wall and its deposition caused by loss-of-function of three RLKs alter root hydrotropism in *Arabidopsis thaliana*

Jinke Chang [1,2], Xiaopeng Li[3], Juan Shen[1], Jun Hu[1], Liangfan Wu[1], Xueyao Zhang[1] & Jia Li [1,2,3] ✉

Root tips can sense moisture gradients and grow into environments with higher water potential. This process is called root hydrotropism. Here, we report three closely related receptor-like kinases (RLKs) that play critical roles in root hydrotropism: *ALTERED ROOT HYDROTROPIC RESPONSE 1* (*ARH1*), *FEI1*, and *FEI2*. Overexpression of these *RLKs* strongly reduce root hydrotropism, but corresponding loss-of-function mutants exhibit an increased hydrotropic response in their roots. All these RLKs show polar localization at the plasma membrane regions in root tips. The biosynthesis of the cell wall, cutin, and wax (CCW) is significantly impaired in root tips of *arh1-2 fei1-C fei2-C*. A series of known CCW mutants also exhibit increased root hydrotropism and reduced osmotic tolerance, similar to the characteristics of the triple mutant. Our results demonstrate that the integrity of the cell wall, cutin, and root cap wax mediate a trade-off between root hydrotropism and osmotic tolerance.

Plants have evolved the ability to grow their roots toward areas with higher water potential in their living environment and thus efficiently obtain water to survive. This fundamental process is called root hydrotropism and was first described in the literature approximately 270 years ago[1,2]. Although the molecular mechanisms controlling this process are poorly understood, significant progress has been made within the last two decades. The root tips, particularly the root caps, play a critical role in the perception of moisture gradients[3–8]. Roots whose caps are removed or covered with hydrophobic materials show a decreased response to a moisture gradient. Numerous studies using Arabidopsis plants have indicated that root bending toward areas with higher water potential does not rely on auxin redistribution[9–12], which is different from the tropism responses observed in plants, including phototropism and gravitropism. Seedlings exhibiting defects in auxin polar transport, either wild-type seedlings treated with auxin polar transport inhibitors in vitro or *pin2* mutants, show enhanced root hydrotropic responses[11–13]. These observations suggest that auxin is not the driving force for the root hydrotropic response and that gravitropism can interfere with root hydrotropism in Arabidopsis.

Takahashi's group identified two nucleotide-substituted Arabidopsis mutants that are not responsive to moisture gradients: *mizu-kussei 1* (*miz1*) and *miz2* (a point mutant of *GNOM*)[14,15]. MIZ1 is a protein of unknown function that contains only a conserved domain of uncharacterized function (the DUF617 domain) and is a soluble protein associated with the cytosolic side of the endoplasmic reticulum (ER) membrane[16]. The detailed molecular mechanisms of MIZ1 in controlling root hydrotropism have not been elucidated. Knocking out *GNOM* leads to a lethal phenotype, whereas the knocking out *MIZ1* does not cause any obvious developmental defects, suggesting that MIZ1 can specifically regulate root hydrotropism, while MIZ2 is involved in multiple biological processes. Using a different screening approach, Cassab's group isolated two mutants with defects in root hydrotropism, which were named *no hydrotropic response 1* (*nhr1*) and *altered hydrotropic response 1* (*ahr1*)[17,18]. However, the corresponding genes for these two mutants have not yet been identified. Thereafter, $Ca^{2+}$, abscisic acid, and brassinosteroids were independently reported to regulate root hydrotropism[13,19–25]. Bennett's group found that Arabidopsis lateral roots can start forming on the wet side of a primary

[1]Ministry of Education Key Laboratory of Cell Activities and Stress Adaptations, School of Life Sciences, Lanzhou University, Lanzhou 730000, China. [2]Gansu Key Laboratory of Gene Editing for Breeding, School of Life Sciences, Lanzhou University, Lanzhou 730000, China. [3]Guangdong Provincial Key Laboratory of Plant Adaptation and Molecular Design, School of Life Sciences, Guangzhou University, Guangzhou 510006, China. ✉e-mail: lijia@gzhu.edu.cn

root[26]. The function of ARF7 is impaired by SUMOylation on the dry side of the primary root. Our group recently revealed that the asymmetric distribution of cytokinins in root tips plays a significant role in root hydrotropism[12]. On the low-moisture-gradient side of an Arabidopsis primary root, more cytokinins accumulate, causing elevated expression of two downstream response regulators, *ARR16* and *ARR17*, which leads to increased cell division activity in the apical meristem.

The plant primary cell wall is composed mainly of cellulose, hemicelluloses, pectin, and a variety of structural proteins[27]. Cellulose microfibrils are the major load-bearing components of the cell wall and are synthesized at the plasma membrane by cellulose synthase (CESA) complexes (CSCs)[28]. Hemicelluloses are a heterogeneous group of polysaccharides that are composed of neither cellulose nor pectin[29]. These polysaccharides generally contain β-(1 → 4)-linked backbones of glucose, mannose, or xylose. Pectins are acidic polysaccharides that are rich in galacturonic acid[30]. Cutin was recently discovered as an outside the cell wall deposition in the root tips of Arabidopsis[31]. It was found that cutin in the root tip plays a vital role in the response to salt stress. It was reported that the integrity of cell wall can be monitored by receptor-like kinases (RLKs), including THE1, FER, WAK, and MIK2[32–35].

RLKs are a group of transmembrane proteins that play significant roles in cell-to-cell and cell-to-environment communications[36]. At least 610 RLKs (including receptor-like cytoplasmic kinases, RLCKs) have been identified in the model plant Arabidopsis[37]. Based on the composition of the extracellular domain, Arabidopsis RLKs can be classified into more than 21 subfamilies, and leucine-rich repeat (LRR)-RLKs belong to the most abundant subfamily. At least 223 LRR-RLKs were found in Arabidopsis[37–39]. To date, only a small fraction of RLKs have been functionally elucidated, and the majority of them still need to be functionally characterized.

Our laboratory is primarily interested in elucidating the biological functions of LRR-RLKs in plants. We previously generated *promoter::GUS* transgenic Arabidopsis plants for all the *LRR-RLKs* and found that the majority of the *LRR-RLKs* were expressed in the roots[39]. We therefore hypothesized that these LRR-RLKs most likely play critical roles in regulating root development and environmental fitness. We screened all our stockpiled *RLK* mutants or overexpressors to test whether they exhibited any altered responses to a moisture gradient. We identified three closely related *RLKs* whose overexpression resulted in a reduced sensitivity to a moisture gradient, and the corresponding loss-of-function mutants exhibited enhanced sensitivity to a moisture gradient. We named one of these three LRR-RLKs Altered Root Hydrotropic Response 1 (ARH1), and two other RLKs were previously designated FEI1 and FEI2[40]. Early studies indicated that double mutants of *FEI1* and *FEI2* exhibited a series of defective phenotypes similar to those of *CESA* mutants, suggesting that FEI1 and FEI2 are involved in the regulation of cell wall biosynthesis[40–42]. Consistently, we found that these three LRR-RLKs are localized to the plasma membrane and are enriched in regions where the cell wall needs to be synthesized or thickened, including new cell plates and the outside layer of the root tip cells. The root tips of the *arh1-2 fei1-C fei2-C* triple mutant exhibited increased sensitivity to a moisture gradient and decreased tolerance to osmotic stress due to defects in the biosynthesis of the cell wall, cutin, and wax (CCW). Other mutants with defects in the CCW exhibited phenotypes similar to those of the triple mutant. Our detailed biochemical and transcriptome analyses revealed that ARH1, FEI1, and FEI2 play key roles not only in the regulation of cell wall biosynthesis but also in its depositions, such as cutin and wax, in the root tips. We demonstrate that CCW plays a trade-off role in the relationship between the root tip hydrotropic response and osmotic tolerance.

## Results

### ARH1, FEI1, and FEI2 mutants exhibit enhanced sensitivity to moisture gradients and reduced osmotic tolerance

To investigate whether RLKs are involved in sensing moisture gradients during root hydrotropism, we tested the hydrotropic response of all our collected *LRR-RLK* mutants and overexpressors on a hydrostimulation medium, a vertically placed split 1/2 MS medium supplemented with D-sorbitol at the bottom right side of the plate (Supplementary Fig. 1). In total, we identified three closely related plasma membrane-localized LRR-RLKs, ARH1, FEI1, and FEI2, and overexpressing their corresponding genes resulted in decreased root hydrotropic curvatures under hydrostimulation treatments, indicating reduced sensitivity to a moisture gradient (Fig. 1a–d and Supplementary Fig. 2). The root tips of two independent *ARH1* single mutants, *arh1-1* and *arh1-2*, showed greater sensitivity to a moisture gradient than did those of Col-0 (Supplementary Fig. 3a, Supplementary Fig. 4c, and Supplementary Fig. 5c). However, the root tips of the *FEI1* or *FEI2* single mutants, *fei1* and *fei2*, did not significantly change in response to a moisture gradient (Supplementary Fig. 3a and Supplementary Fig. 4a, b). We subsequently attempted to generate double and triple mutants using T-DNA-inserted null single mutants of these three *RLKs* by genetic crossing. We obtained two double mutants, *arh1-1 fei1* and *arh1-1 fei2*. We also isolated *arh1-1 fei1 fei2(+/−)*, in which *arh1-1* and *fei1* were homozygous and *fei2* was heterozygous. The hydrotropic responses of the root tips of these mutants were not significantly different from those of *arh1-1* (Supplementary Fig. 4). These data suggest the functional redundancy of these three LRR-RLKs, especially between FEI1 and FEI2. Similarly, the amino acid identity between FEI1 and FEI2 reached 82.06%, whereas the amino acid identities between ARH1 and FEI1 and between ARH1 and FEI2 were 49.66% and 48.31%, respectively. Unfortunately, we failed to obtain a homozygous triple mutant in the offspring of *arh1-1 fei1 fei2(+/−)*, suggesting that complete knockout of these three genes in a single plant likely resulted in a lethal phenotype.

To successfully generate triple mutants of *ARH1*, *FEI1*, and *FEI2*, we used a CRISPR–Cas9 system to partially delete *FEI1* and *FEI2* in the *arh1-2* background (Supplementary Fig. 3b, c). We genotyped more than one thousand independent lines and obtained more than 20 independent triple mutant lines in which both *FEI1* and *FEI2* were edited, and most of these lines failed to generate homozygous lines for both genes. Only four independent lines produced homozygous offspring. Sequence analyses revealed that at least one of the *FEI1* and *FEI2* genes exhibited editing events in the promoter or noncoding sequence regions in three out of the four independent triple-mutant lines, suggesting that their corresponding proteins exhibited a potentially nonedited status. One triple mutant, named *arh1-2 fei1-C fei2-C*, contained deletions in the coding regions of both *FEI1* and *FEI2*, which possibly resulted in a null allele of *FEI1* and a partially dysfunctional FEI2 (Supplementary Fig. 3b, c). The triple mutant *arh1-2 fei1-C fei2-C* was backcrossed with Col-0 to obtain a series of single and double mutants of *arh1-2*, *fei1-C*, and *fei2-C*, and the triple mutants with CRISPR–Cas9 constructs were segregated. The roots of the double and triple mutants generated by CRISPR–Cas9 showed significantly greater sensitivity to a moisture gradient than did those of the wild-type Col-0 plants (Fig. 1e–h, Supplementary Fig. 5, Supplementary Fig. 6a, b, and Supplementary Movie 1). The gravitropism of the triple mutant was similar to that of the wild-type under both normal and osmotic stress conditions (Supplementary Fig. 6c, d, Supplementary Fig. 7, and Supplementary Movie 2). With the exception of having slightly thicker roots than that of the wild-type, the triple mutant presented no additional developmental defects under normal growth conditions (Supplementary Fig. 8 and Supplementary Fig. 9a, e, i, m). The root tips of the triple mutant were swollen after the plants were transferred from normal 1/2 MS media to split 1/2 MS media supplemented with 800 mM D-sorbitol on the bottom right side of the media or to osmotic stress media supplemented with 400 mM D-sorbitol (Fig. 1g and Supplementary Fig. 9). Consistent with the findings of our previous study, we found that the difference in the cortical cell number between the sides of the root tips with lower and higher water potential was greater in *arh1-2 fei1-C fei2-C* than in Col-0 after hydrostimulation treatment (Fig. 1i–o)[12].

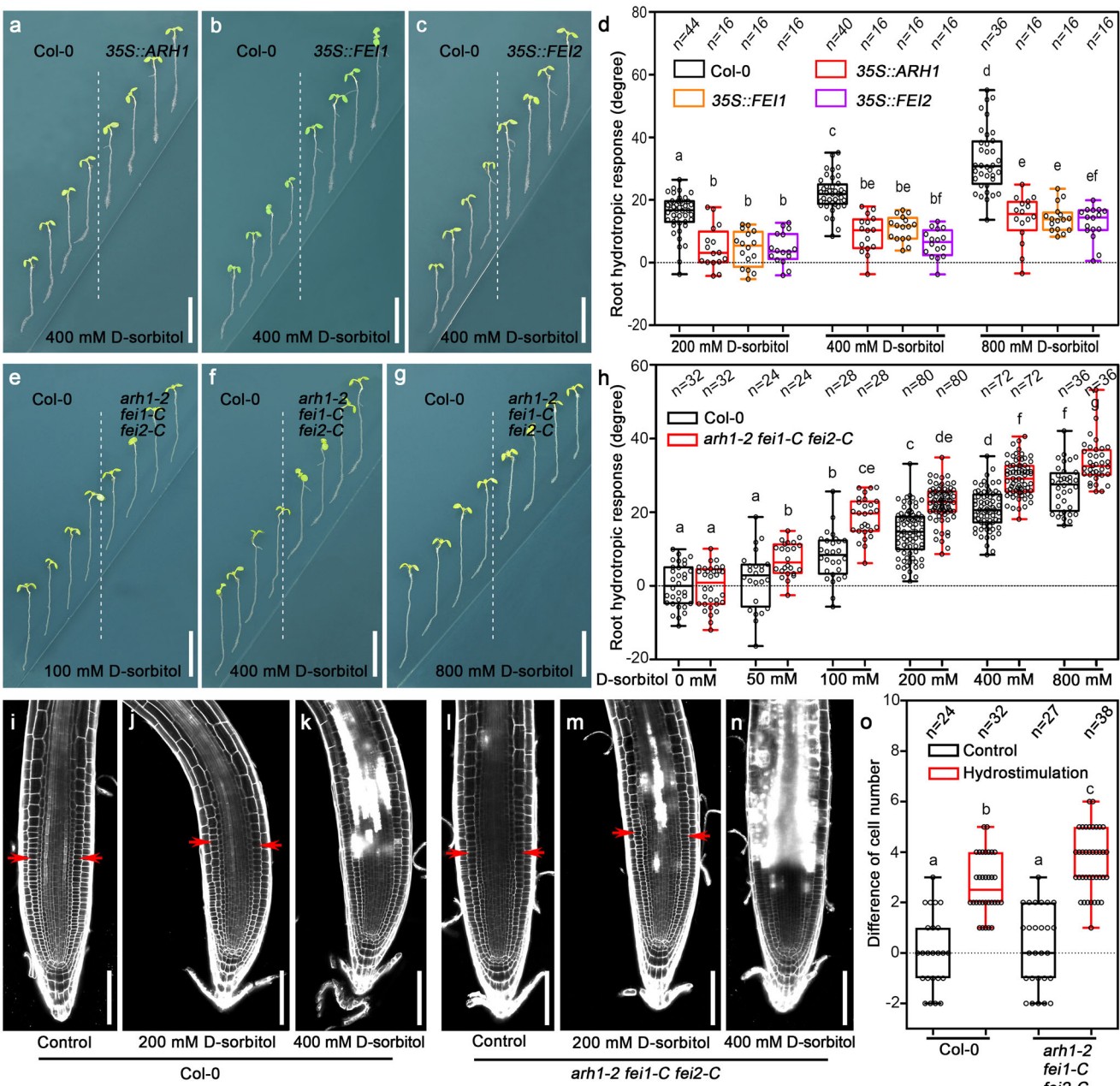

**Fig. 1 | Three closely related RLKs, including ARH1, FEI1, and FEI2, negatively regulate root hydrotropism. a–c** Transgenic plants overexpressing *ARH1, FEI1, or FEI2* show decreased root hydrotropic response. **d** Measurements of root hydrotropic curvatures of Col-0, *35 S::ARH1, 35 S::FEI1*, and *35 S::FEI2* transgenic seedlings. **e–g** Root hydrotropic response of wild-type (Col-0) and the triple mutant (*arh1-2 fei1-C fei2-C*). **h** Measurements of root hydrotropic curvatures. **i–n** Propidium iodide (PI)-stained root tips of Col-0 and the triple mutant after 2-h hydrostimulation treatments. **o** Cortex cell number differences of right side versus left side (control) for Col-0 and the triple mutant as represented in (**i, l**), and lower water potential side versus higher water potential side (hydrostimulation) as represented in (**j**) and (**m**) within a 200 μm meristematic zone starting from the quiescent center. Red arrow heads in (**i, j, l, m**) indicate the borders of 200 μm meristem zone starting from the quiescent center. Boxplots span the first to the third quartiles of the data, and whiskers indicate the minimum and maximum values. The line in the box represents the mean. Each circle represents the measurement of an individual root tip (**d, h, o**). "*n*" represents the number of roots analyzed in the experiment. Three independent biological replicates were carried out. Scale bars represent 10 mm in (**a–c, e–g**) and 100 μm in (**i–n**). One-way ANOVA with Tukey's multiple comparison test was used for statistical analyses with $P < 0.01$.

In addition to the altered hydrotropic response, the root tips of single, double, and triple mutant plants harboring these three *LRR-RLKs* also exhibited decreased osmotic tolerance compared with Col-0 plants, as indicated by an expanded area of dead cells, which was revealed by propidium iodide staining after hydrostimulation or osmotic stress treatment (Fig. 1k, m, n, Supplementary Fig. 10, and Supplementary Fig. 11). Conversely, the root tips of the overexpressors harboring *35S::ARH1, 35S::FEI1*, and *35S::FEI2* showed enhanced tolerance to osmotic stress in comparison with the wild type (Supplementary Fig. 12). The root tip defects of *arh1-2 fei1-C fei2-C* can be complemented by the expression of *pARH1::gARH1-YFP, pFEI1::gFEI1-YFP*, or *pFEI2::gFEI2-YFP* (Supplementary Fig. 13a–e and Supplementary Fig. 14). In summary, our genetic data indicated that the phenotype of *arh1-2 fei1-C fei2-C* was indeed caused by mutations in these three *LRR-RLKs*. We therefore conducted our subsequent analyses using this special mutant, *arh1-2 fei1-C fei2-C*.

## ARH1, FEI1, and FEI2 are polarly localized in primary root tip cells

To elucidate the functions of ARH1, FEI1, and FEI2 in regulating root hydrotropism, we first analyzed the subcellular localization and distribution of these proteins in the root tips of Arabidopsis plants. We generated transgenic lines harboring *pARH1::gARH1-YFP*, *pFEI1::gFEI1-YFP*, or *pFEI2::gFEI2-YFP* in Col-0 and *arh1-2 fei1-C fei2-C*. Interestingly, all three YFP-fused proteins exhibited polar subcellular localization on the plasma membrane in the root tip cells (Fig. 2 and Supplementary Fig. 13f–h). ARH1-YFP is mainly localized on the plasma membrane facing the newly formed cell plate during cell division. However, FEI1-YFP and FEI2-YFP are enriched in the plasma membrane of epidermal cells, facing the root surface. The polar localizations of ARH1-YFP, FEI1-YFP, and FEI2-YFP were not altered by hydrostimulation treatment (Supplementary Fig. 15a–i). The newly formed cell plate and outer domain of the root surface cells are the places where cell wall biosynthesis and deposition usually occur, as supported by data from previous studies demonstrating the involvement of FEI1 and FEI2 in this process[40]. We therefore hypothesized that the altered root hydrotropism and osmotic tolerance of the triple mutant may be attributed to impairments in cell wall biosynthesis or thickening. The number of newly formed cell plates in the meristematic cortex was clearly increased on the lower water-potential side, as indicated by the accumulation of ARH1-YFP (Supplementary Fig. 15j, k). This observation is consistent with one of our previous discoveries, suggesting that cell division activity on the lower water-potential side is greater than that on the higher water-potential side in the root meristematic region[12].

## The CCW is largely damaged in the root tips of *arh1-2 fei1-C fei2-C* plants

The cell wall is mainly composed of cellulose, hemicellulose, and pectin and is further covered by the deposition of cutin on the outside layer of the root surface, which was discovered a few years ago[31]. We extracted the cell wall from the root tips of Col-0 and *arh1-2 fei1-C fei2-C* plants and analyzed the principal components of these plants. We found that the levels of cellulose and galacturonic acid (a main component of pectin) in the root tips of *arh1-2 fei1-C fei2-C* were significantly lower than those in the root tips of Col-0 (Fig. 3a, b and Supplementary Fig. 16a)[43]. Pectin methylation was also altered in the root tips of *arh1-2 fei1-C fei2-C* plants (Supplementary Fig. 16b–d).

A reduction of the pectin content in the root tips and seed coat mucilage of *arh1-2 fei1-C fei2-C* were further confirmed by ruthenium red staining, a pectin-specific dye (Fig. 3c–e and Supplementary Fig. 17)[44,45]. We also visualized cutin in the root tips of Col-0 and *arh1-2 fei1-C fei2-C* via fluorol yellow (FY) staining and transmission electron microscopy (TEM) analyses (Fig. 3f–t)[46,47]. The cutin layer of *arh1-2 fei1-C fei2-C* was markedly thinner than that of Col-0. A *gso1 gso2* double mutant was used as a negative control because it barely synthesizes cutin[31,48]. Conversely, the amounts of cellulose, pectin, and cutin were highly increased in the root tips of the *ARH1*-, *FEI1*-, and *FEI2*-overexpression plants (Supplementary Fig. 18).

Surprisingly, we found a layer of electron-transparent cell wall deposition covering the cutin in the root caps of Col-0 plants and assumed that it was wax. This assumption was confirmed using the *kcs7 kcs15 kcs20 kcs21* quadruple mutant, in which the wax biosynthesis pathway was blocked, as a negative control. This layer was no longer observed in the quadruple mutant (Fig. 3l, p, t)[49]. The disappearance of this layer in *arh1-2 fei1-C fei2-C* suggested that these three RLKs are involved in the regulation of wax biosynthesis in the root caps (Fig. 3j, n, r). The compositions and amounts of wax and cutin covering the root tips of Col-0, *arh1-2 fei1-C fei2-C*, overexpressors of *ARH1*, *FEI1*, and *FEI2*, *gso1 gso2*, and *kcs7 kcs15 kcs20 kcs21* were analyzed by GC–MS. The amounts of root tip wax and cutin were significantly decreased in *arh1-2 fei1-C fei2-C* and increased in the overexpressors of the three *LRR-RLKs* (Fig. 4 and Supplementary Fig. 19). Our analyses indicated that root cap wax is mainly composed of alkanes, alcohols, ketones, and fatty acids. The most abundant component of the root tip wax was alkanes. The carbon chain lengths of the detected alkanes ranged from C16 to C34. C29 ketone and C16 to C20 fatty acids were detected in the root tips, and their significant reductions were found in *arh1-2 fei1-C fei2-C* relative to Col-0. Our TEM analysis indicated that the root cap wax was removed following wax extraction. We also observed that root cap wax accumulated as droplets in the outermost layer of the root cap cells and was subsequently transported into the outer layer of the cell wall (Supplementary Fig. 20).

## The expression of genes involved in the biosynthesis of CCW is strongly downregulated in the root tips of *arh1-2 fei1-C fei2-C*

Our analyses revealed that the composition and amount of CCW in *arh1-2 fei1-C fei2-C* are greatly altered. To understand the molecular

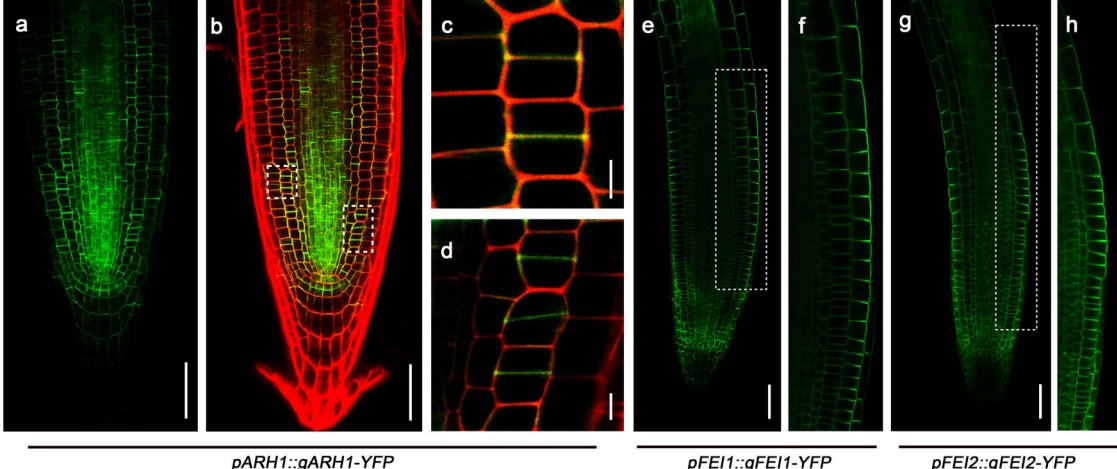

**Fig. 2 | ARH1, FEI1, and FEI2 proteins show polar localization in the cells of primary root tips. a** ARH1-YFP from *pARH1::gARH1-YFP* transgenic plants can be visualized in the primary root meristematic zone under a confocal microscope. **b** YFP signal of ARH1-YFP were merged with its corresponding propidium iodide-stained root tip picture. **c, d** Magnifications of partial areas in figure (**b**), as marked in dashed rectangular boxes. **e, g** FEI1-YFP from *pFEI1::gFEI1-YFP* transgenic plants and FEI2-YFP from *pFEI2::gFEI2-YFP* transgenic plants can be visualized in the primary roots and show clear polar localization at the root surface under a confocal microscope. **f, h** Magnifications of the rectangular boxes shown in figures (**e, g**), respectively. Scale bars represent 50 μm in (**a, b, e, f**), and 5 μm in (**c, d**). Five biological replicates were performed.

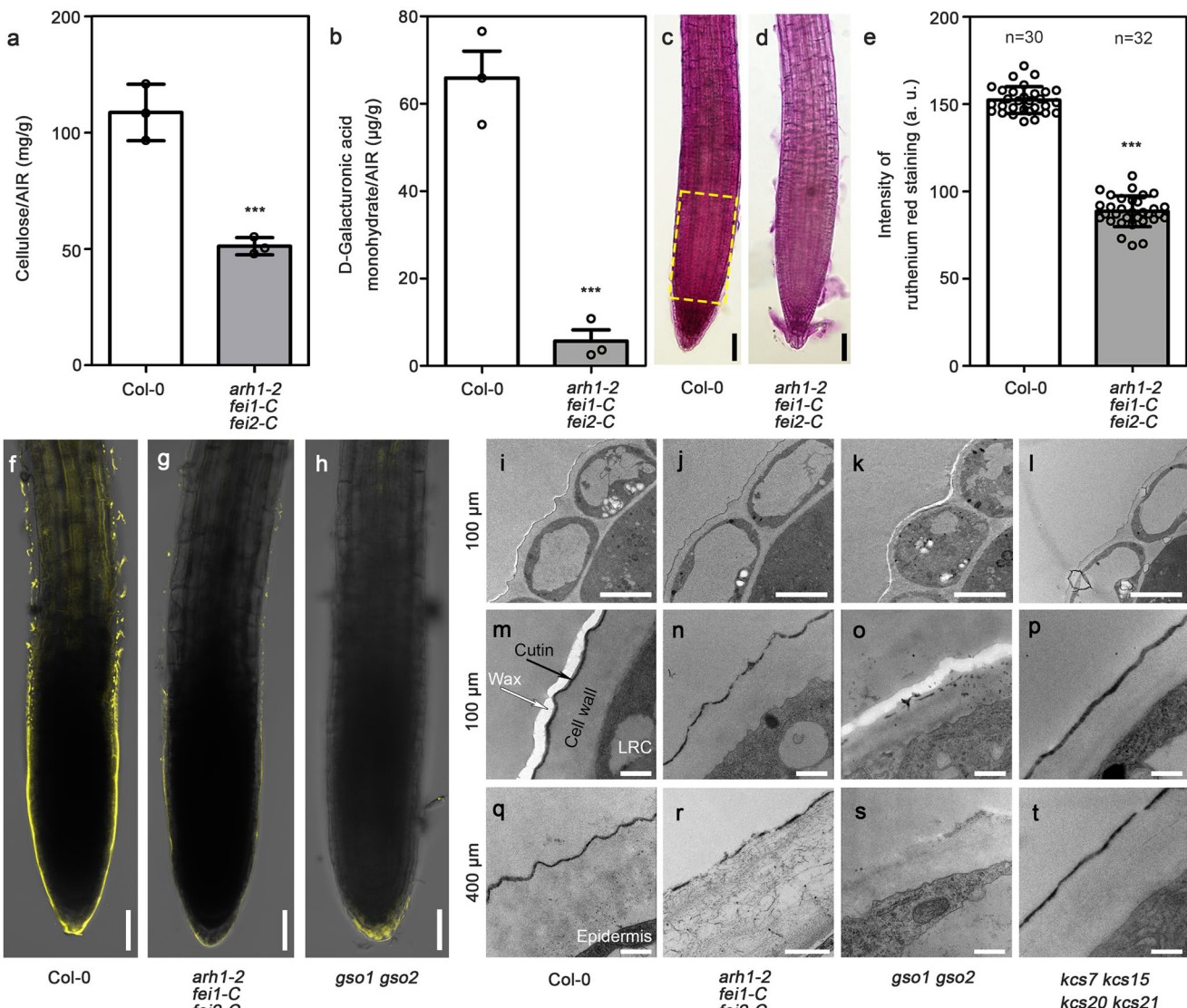

**Fig. 3 | The root tips of *arh1-2 fei1-C fei2-C* show defects in CCW. a** The amount of cellulose in the cell wall of the root tips of Col-0 and the triple mutant was quantified. The total cell wall was presented as alcohol-insoluble residues (AIR). **b** D-Galacturonic acid monohydrate was measured in cell wall (AIR) of Col-0 and the triple mutant root tips by GC-MS. Ruthenium red stained root tips from four-day-old Col-0 (**c**) and the triple mutant (**d**) seedlings. **e** Relative staining intensity for the root tips as represented in (**c**), (**d**) in a 200 μm × 100 μm area (as shown in **c**). **f–h**, Fluorol yellow (FY) stained root tips from seedlings of two-day-old Col-0, *arh1-2 fei1-C fei2-C*, and *gso1 gso2* mutants. **i–t** Transmission electron microscope (TEM) images showing CCW of the outermost lateral root cap cells (100 μm from the root tip) (i-p) and the outermost epidermal cells in the elongation zone (400 μm from

the root tip) (**q–t**) of two-day-old Col-0, *arh1-2 fei1-C fei2-C*, *gso1 gso2*, and *kcs7 kcs15 kcs20 kcs21* quadruple mutants. LRC represents lateral root cap. Data are the means ± SD of three biological replicates (**a**, **b**). Each circle represents a measurement of a biologically independent sample (**a**, **b**) or an individual root tip (**e**). "*n*" represents the number of roots analyzed in the experiment (**e**). Three biological replicates were carried out for ruthenium red staining and YF staining experiments, while ten biological replicates were carried out for TEM analyses. Scale bars represent 50 μm in (**c**, **d**, **f–h**), 3 μm in (**i–l**), and 300 nm in (**m–t**). Statistical significance was determined by two-side and unpaired *t*-test, without making any adjustments for multiple comparisons (*P* < 0.01).

mechanisms causing these defects, we compared the gene expression profiles of the root tips of Col-0 and *arh1-2 fei1-C fei2-C*. We extracted total RNA from 5-mm-long root tips of Col-0 and *arh1-2 fei1-C fei2-C* plants and conducted RNA-seq analyses. In total, we identified 622 differentially expressed genes (DEGs) in *arh1-2 fei1-C fei2-C* compared with Col-0. Kyoto Encyclopedia of Genes and Genomes (KEGG) pathway analysis indicated that these DEGs were enriched mainly in the biosynthetic and catabolic pathways of fatty acids and sugar metabolism, all of which are related to the biosynthesis of cutin and wax (Fig. 5a). We next performed RT–qPCRs for 102 genes known or predicted to encode enzymes catalyzing the biosynthesis of the cell wall, fatty acids, cutin, or wax[30,50–53]. The majority of these genes were significantly downregulated in the root tips of *arh1-2 fei1-C fei2-C* and in

the roots of various single and double mutants of *arh1-2*, *fei1-C*, and *fei2-C* (Fig. 5b–e and Supplementary Fig. 21a). Similar results were observed in another independent set of single and double mutants (T-DNA inserted alleles), *arh1-1*, *fei1*, and *fei2*. But these genes were upregulated in the *ARH1-*, *FEI1-* or *FEI2*-overexpression lines (Supplementary Fig. 21b, c). The reduced expression of these genes in *arh1-2 fei1-C fei2-C* could be largely rescued by the introduction of *pARH1::-gARH1-YFP*, *pFEI1::gFEI1-YFP*, or *pFEI2::gFEI2-YFP* (Supplementary Fig. 21d). Several genes encoding expansins (EXPAs), a group of nonenzymatic cell wall active proteins that can relax the plant cell wall, also exhibited decreased expression in root tips of the triple mutant (Fig. 5f)[54]. These results indicated that ARH1, FEI1, and FEI2 are involved in the regulation of CCW biosynthesis. In addition, an analysis of the

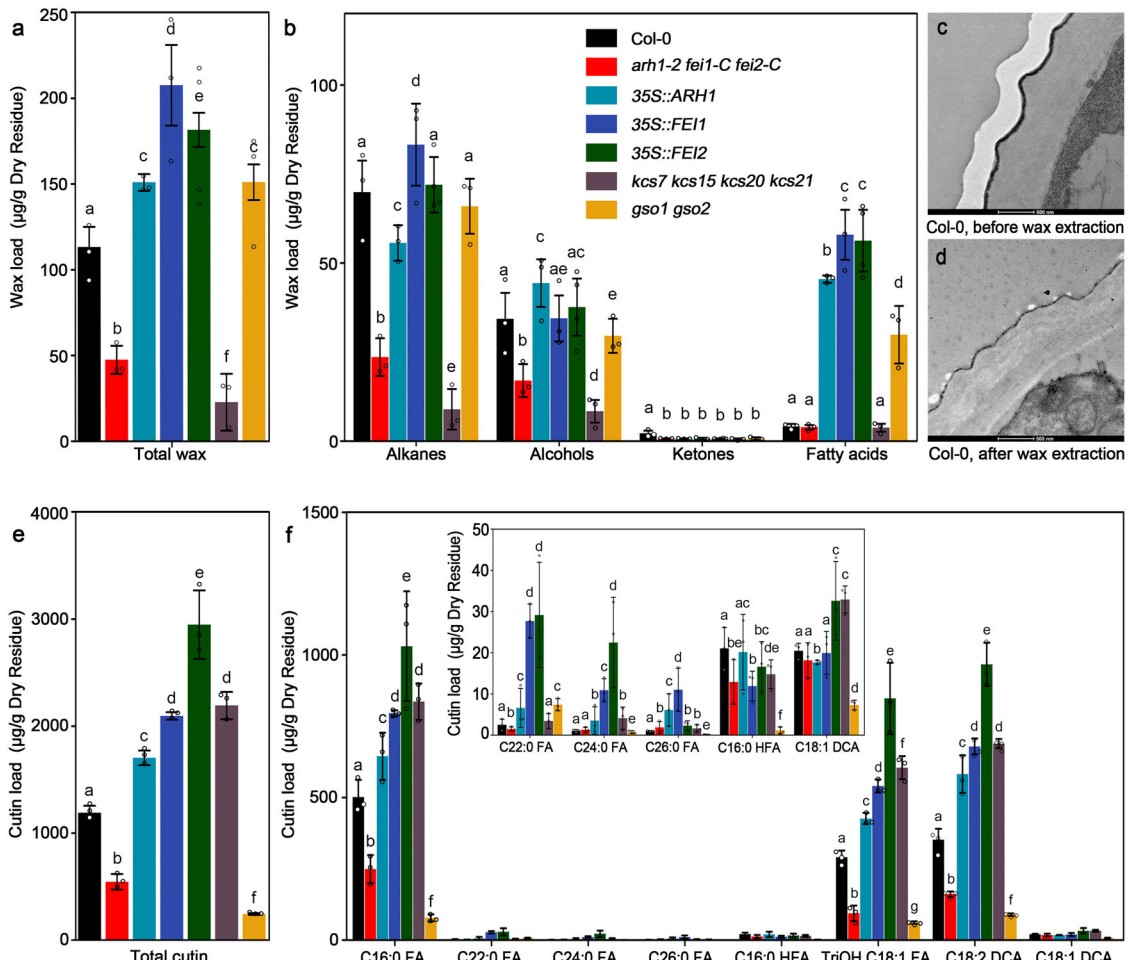

**Fig. 4 | The amount of total and each chemical composition of wax and cutin in the root tips of wild-type, overexpressors, and various mutants. a** The amount of total wax in different genotypes. **b** The amount of each chemical composition of wax in different genotypes. **c**, **d** TEM images showing CCW at the surface of the outermost lateral root cap cells (100 μm from the root tip) before and after wax extraction. Three biological replicates were carried out. **e** The amount of total cutin in various genotypes. **f** Amount of each chemical composition of the cutin in various genotypes. The measurements were carried out by using gas chromatography-mass spectrometer (GC-MS) (**a**, **b**, **e**, **f**). FA fatty acids, HFA ω-hydroxy fatty acids, TriOH 18:1 FA 9,10,18-triOH C18:1 fatty acids, DCA dicarboxylic acids. Each circle represents a measurement of a biologically independent sample. Data are the means ± SD of three biological replicates (**a**, **b**, **e**, **f**). Scale bars represent 500 nm in (**c**, **d**). One-way ANOVA with Tukey's multiple comparison test was used for statistical analyses with $P < 0.01$.

expression levels of numerous FEI2 regulated genes (genes that exhibited downregulation in the *fei2-C* and *fei2* mutants) confirmed that *fei2-C* is a weak mutant of *FEI2* (Supplementary Fig. 21a, b and Supplementary Fig. 22). *fei2-C* encodes a protein with a 70-amino-acid deletion in the extracellular domain of FEI2 (Supplementary Fig. 3c).

Fatty acids are not only precursors of cutin and wax but also building blocks of lipids[53,55]. We hypothesized that lipid metabolism should also be altered in *arh1-2 fei1-C fei2-C*. We therefore applied a lipidomic approach to analyze the lipid composition and amount in the root tips of Col-0 and *arh1-2 fei1-C fei2-C* and found strong alterations (Supplementary Fig. 23). Because lipids are the critical components of the cell membrane, these data suggest that the three LRR-RLKs may also regulate the biosynthesis of the cell membrane. The integrity of the cell membrane could also be altered in *arh1-2 fei1-C fei2-C*.

**Mutants with defects in CCW all exhibit enhanced sensitivity to moisture gradients and reduced osmotic tolerance**

Our results demonstrated that the biosynthesis of CCW is significantly impaired in the mutants of *ARH1*, *FEI1*, and *FEI2*. The subsequent question is whether the increased sensitivity to moisture gradients and decreased osmotic tolerance of *arh1-2 fei1-C fei2-C* are the consequence of reduced integrity of the CCW at the root tips. We analyzed numerous mutants known to have defects in the biosynthesis of CCW. For example, *cesa1* and *cesa6* are single-amino-acid-substitution mutants of cellulose synthetase. Indeed, the level of cellulose in the cell wall of *cesa6* roots is highly reduced (Supplementary Fig. 18a)[56]. In addition, *gso1 gso2*, a double mutant of two closely related *LRR-RLKs*, and a single mutant of *BDG*, *bdg*, have been shown to have no cutin in Arabidopsis root tips[31]. *BDG* encodes an α/β-hydrolase family protein that localizes to the outermost portion of the epidermal cell wall. We demonstrated that *kcs7 kcs15 kcs20 kcs21* is a quadruple mutant with defects in root cap wax (Figs. 3n, p, t, and 4). Our analyses indicated that the root tips of all these mutants exhibited enhanced sensitivity to moisture gradients and decreased osmotic tolerance compared with those of Col-0 (Fig. 6a–c, f, Supplementary Fig. 24, and Supplementary Fig. 25). We applied pectinase, an enzyme that can efficiently degrade pectin in vitro, to reduce the level of pectin in the root tips of Col-0 plants (Supplementary Fig. 26a, b). After treated with pectinase, the root tips showed enhanced sensitivity to moisture gradients and decreased osmotic tolerance compared with those of the control (Fig. 6d, e, g and Supplementary Fig. 26c–g). These results provide genetic and physiological evidence indicating that defects in the biosynthesis of CCW in the *arh1-2 fei1-C fei2-C* triple mutant are

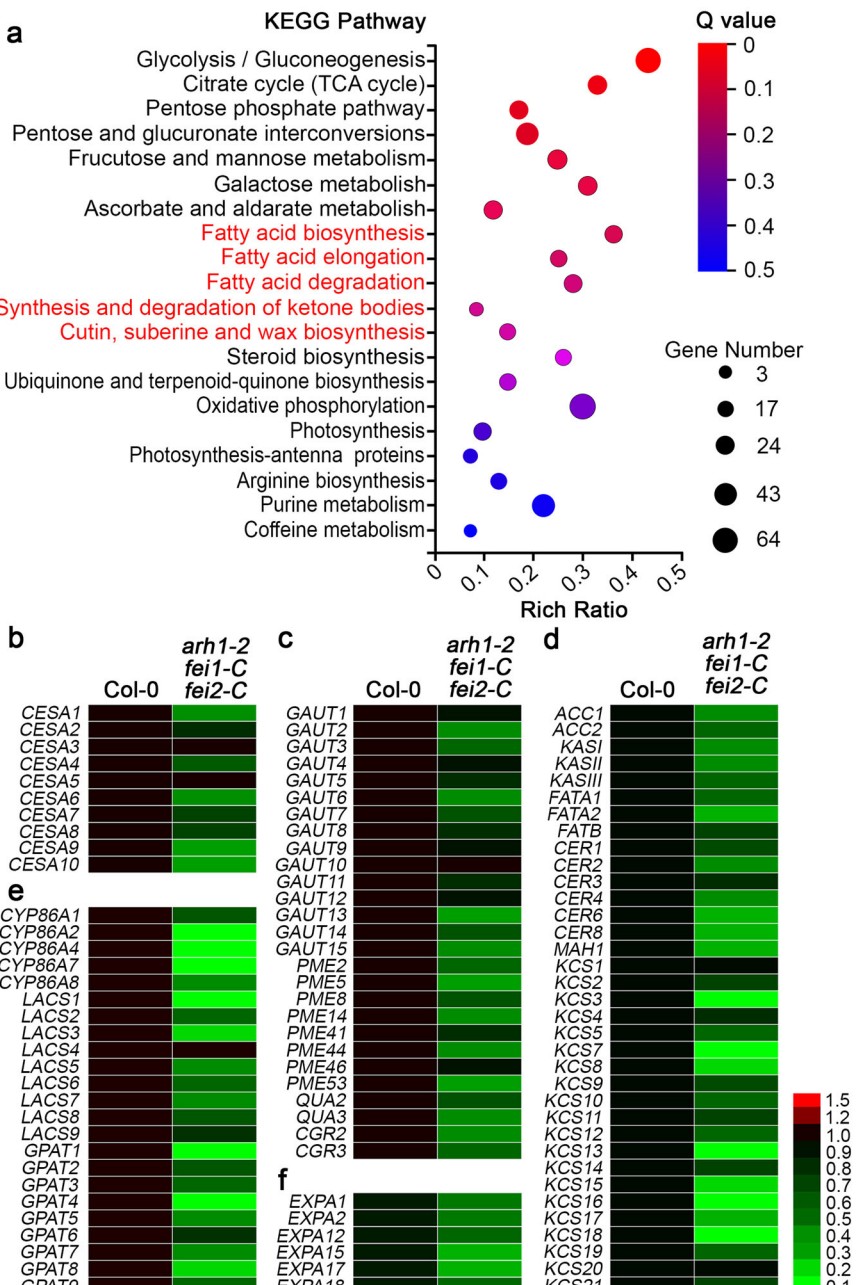

**Fig. 5 | Transcriptional levels of most genes coding for the enzymes catalyzing the biosynthesis of CCW are downregulated in root tips of the triple mutant compared with those in Col-0. a** Kyoto Encyclopedia of Genes and Genomes (KEGG) pathways of differentially expressed genes (DEGs) in root tips of *arh1-2 fei1-C fei2-C* compared with those in Col-0 identified by RNA-seq analysis. KEGG pathways were analyzed using KEGG Mapper. The pathways directly related to cutin and wax were marked in red. Source data for RNA-seq are included in the Source Data file. The "Rich Ratio" represents the enrichment degree of DEGs in each pathway. The number of the enriched DEGs is represented by the size of each circle, with a larger circle indicating a greater number. The Q value represents the adjusted *p* value after multiple testing, which was completed by using Bonferroni correction, and a smaller Q value indicates a more significant enrichment effect. RT-qPCR analyses confirm the expression levels of most genes coding for the biosynthesis enzymes of cell wall (**b, c**), cutin, and wax (**d, e**), and cell wall expansion (**f**) are significantly downregulated in root tips of *arh1-2 fei1-C fei2-C* compared with those in Col-0. Data are the mean of three biological replicates.

directly involved in the hypersensitivity of the root tips to moisture gradients and reduced osmotic tolerance.

## Discussion

Our detailed genetic, cell biology, biochemical, and physiological experiments indicated that the biosynthesis of CCW plays key roles in regulating the response of the root tips to moisture gradients and osmotic tolerance. First, single, double, and triple mutants of *ARH1*, *FEI1*, and *FEI2*, generated via different approaches, all exhibited enhanced sensitivity to moisture gradients and decreased tolerance to osmotic stress, displaying an increased hydrotropic bending curvature and an extended area of cell death upon exposure to osmotic stress. Second, the cell wall and its depositions including cutin and wax (CCW), were significantly damaged in the root tips of the triple mutant. Third, the expression of genes known to encode enzymes catalyzing the biosynthesis of CCW were significantly downregulated in the root tips of *arh1-2 fei1-C fei2-C*. Fourth, other known CCW mutants with defects also exhibited enhanced responses to moisture gradients and decreased osmotic tolerance, similar to the *arh1-2 fei1-C fei2-C* triple mutant.

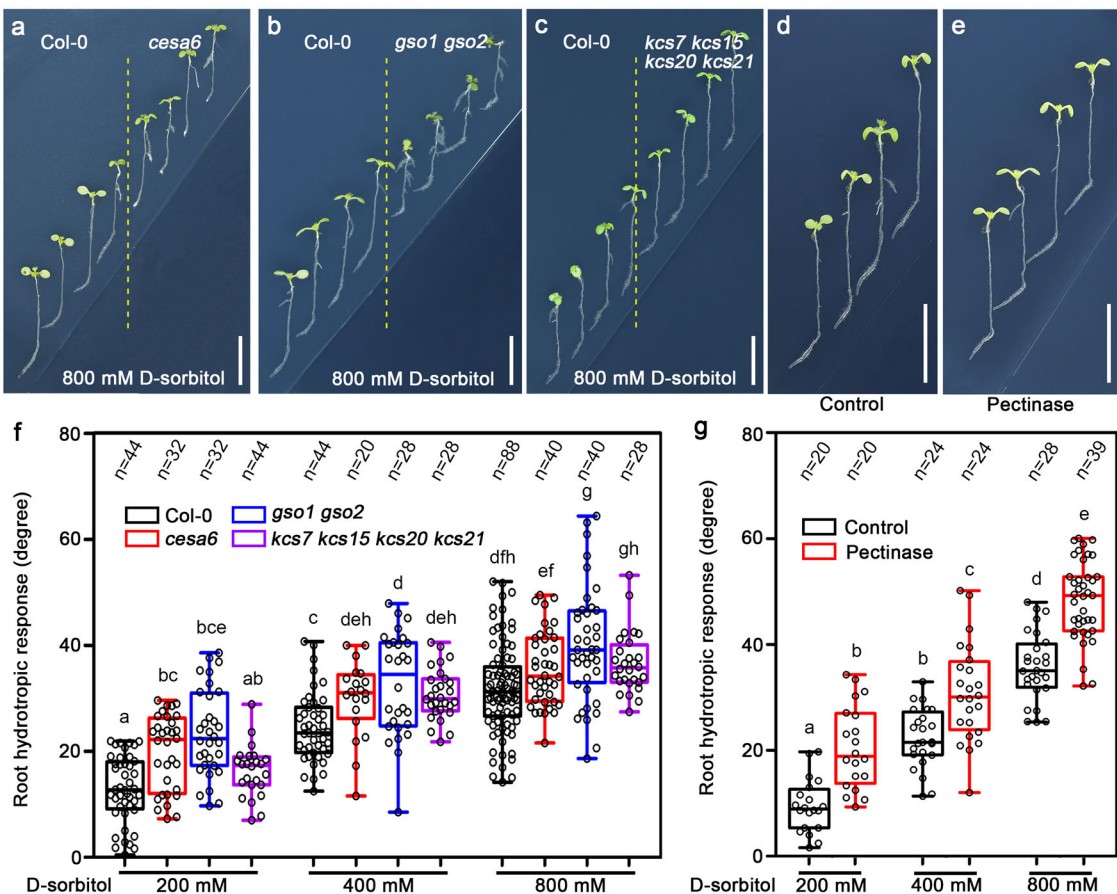

**Fig. 6 | Mutants defective in the biosynthesis of CCW show enhanced response to moisture gradients. a–c** Hydrotropic responses of wild-type (Col-0) and various defective mutants of CCW. **d, e** Hydrotropic response of Col-0 seedlings treated with 800 mM D-sorbitol at the bottom right side, supplemented with DMSO as a control (**d**) or 50 μM pectinase (**e**) at the both sides of the split medium. **f** Measurements of the root growth curvatures of *cesa6* single, *gso1 gso2* double, and *kcs7 kcs15 kcs20 kcs21* quadrupole mutants after 24-h hydrostimulation treatments. **g** Measurements of the root hydrotropic curvatures of Col-0 treated with control or pectinase. Boxplots span the first to the third quartiles of the data, and whiskers indicate the minimum and maximum values. The line in the box represents the mean. Each circle represents the measurement of an individual root (**f, g**). "*n*" represents the number of roots analyzed in the experiment. Three biological replicates were carried out. Scale bars represent 10 mm in (**a–e**). One-way ANOVA with Tukey's multiple comparison test was used for statistical analyses with $P < 0.01$.

A previous study indicated that the *fei1 fei2* double mutant displays a swollen root tip phenotype when transferred from 1/2 MS medium supplemented with 1% sucrose to 1/2 MS medium supplemented with 4.5% sucrose[40]. This phenotype is similar to that of a cellulose synthase mutant, *cesa6*, or to that of a wild-type plant treated with the cellulose biosynthesis inhibitor isoxaben[41]. A mutant of the fasciclin-like GPI-anchored extracellular arabinogalactan protein SOS5, *sos5*, exhibited a swollen root tip phenotype under high-concentration sucrose treatment, similar to *fei1 fei2*[40,41,57]. The *fei1 fei2 sos5* triple mutant did not exhibit additive root tip defects on 1/2 MS media supplemented with 4.5% sucrose, suggesting that FEI1, FEI2, and SOS5 regulate the response of the root tip to high concentrations of sucrose, likely through the same signaling pathway. Additionally, a *fei2* single mutant exhibited a defect in seed mucilage adherence[42]. The seed mucilage is composed mainly of pectin, which is an important component of the cell wall. Previous studies indicated that FEI1 and FEI2 are involved in the regulation of cell wall biosynthesis; however, whether they are also involved in the regulation of the biosynthesis of cell wall depositions has not been determined. The molecular mechanisms underlying the regulation of cell wall biosynthesis by these two RLKs have not been fully elucidated. Our comprehensive analyses indicated that cell wall components, such as cellulose and pectin, as well as cell wall depositions, including cutin and wax, are significantly decreased in the *arh1-2 fei1-C fei2-C* triple mutant but increased in the *ARH1-*, *FEI1-*,

or *FEI2*-overexpression lines (Fig. 3, Supplementary Figs. 18, and 19). We also analyzed the expression levels of genes whose encoded proteins were confirmed or predicted to catalyze the biosynthesis of cellulose, pectin, cutin, and wax. The majority of these genes were strongly downregulated in the root tips of the triple mutant but upregulated in the overexpression plants (Fig. 5 and Supplementary Fig. 21). Our data provide strong evidence demonstrating that these three RLKs are involved in the regulation of not only the biosynthesis of the cell wall but also its depositions. It is likely that the ARH1-mediated signaling pathway controls the expression of genes in the CCW biosynthesis pathways via one or a group of key transcription factors yet to be identified. The identification of key transcription factors relies on future elucidation of the ARH1-mediated signaling pathway, including the ligands of ARH1 and downstream regulatory components.

Primary cell wall depositions are widely observed in the aerial parts of plants. Cell wall depositions are predominantly composed of fatty acid-derived polyesters located outside the cell wall to protect plant cells from desiccation, mechanical damage, and pathogen attacks[43,56]. Such depositions have seldomly been studied in the roots until recently. It was reported that Arabidopsis root caps are covered by a layer of electron-opaque material named cutin[31]. Through TEM analysis, we found that cutin covers the root surface not only in the root cap region but also in some regions of the elongation zone.

Interestingly, we also detected a layer of electron-transparent cell wall deposition outside the cutin, which was identified only in the root cap region and not in the elongation zone (Fig. 3i–t). Through mutant analysis and composition determination via GC–MS, we confirmed that this layer of deposition was actually wax. To our knowledge, the presence of root cap wax has not been previously reported in the literature. Our analyses indicated that the root cap cutin and wax play key roles in regulating the sensitivity of the root tip to moisture gradients and osmotic stress.

The CCW complex acts as a protective barrier against osmotic stress in root tips. The permeability of cells, as revealed by toluidine blue staining, was significantly enhanced in the root tips with defects in the CCW. Conversely, the permeability of cells was decreased in the overexpression plants (Supplementary Fig. 27). The root tip CCW apparently plays key roles in the trade-off between the hydrotropic response and osmotic tolerance (Supplementary Fig. 28). On the one hand, a thicker CCW may increase the toughness of cells against osmotic stress, and on the other hand, we found that the mutants with defects in the biosynthesis of CCW exhibited increased sensitivity to a moisture gradient. In the book "*The power of movement in plants*" (1880, p180–186), Darwin and Darwin found that root tips play a critical role in perceiving moisture gradients[3]. In their experiments, root tips (1.5–2 mm in length) of *Phaseolus multiflorus*, *Vicia faba*, and *Avena sativa* were coated with a mixture of olive oil and lamp black, and the hydrophobic material-coated root tips exhibited decreased sensitivity to a moisture gradient. The root tip CCW, which includes hydrophobic cutin and wax, serves as a natural barrier to water. These compounds play a similar role as the mixture of olive oil and lamp black used by Darwin and Darwin in preventing water loss and maintaining normal cell turgor under moisture gradient or osmotic stress conditions. This process is crucial for sustaining cellular activities such as cell division in the root meristem region. Interestingly, the triple mutant displayed clearly increased meristematic cortex cell numbers compared with the wild-type (Supplementary Fig. 10j). In addition, the difference in the cell division rate between the sides of the root tips with lower and higher water potential was higher in the triple mutant than in the wild-type. These results are consistent with the findings from one of our previous studies, which showed that the asymmetric distribution of cytokinins and the uneven rate of cell division between the sides of the root tips with lower and higher water potential are critical to the hydrotropic response of roots[12]. Whether the integrity of the CCW directly or indirectly affects cell division is an outstanding research avenue for future investigations. These studies help characterize the root hydrotropic response and provide possible future strategies for creating optimal crops with higher sensitivity to moisture gradients and stronger tolerance to osmotic stress.

## Methods

### Plant materials and growth conditions
The Columbia accession (Col-0) was used as the wild type. T-DNA insertion mutants of *ARH1* (*At5g62710*), *arh1-1* (SAIL_119_A09), *arh1-2* (SALK_085175), *fei1* (SALK_080073), *fei2* (SALK_044226), and *bdg* (SALK_208738C) were obtained from the Arabidopsis Biological Resource Center (ABRC). The *arh1-2 fei1-C fei2-C* triple mutant was generated using a CRISPR–Cas9 system to edit *FEI1* and *FEI2* in the *arh1-2* background. *cesa6* is a single nucleotide mutation (T2685G) with an early stop codon generated by our laboratory. The *gso1 gso2* double mutant was previously described[48], and the quadruple mutant *kcs7 kcs15 kcs20 kcs21* was kindly provided by Zhongnan Yang (Shanghai Normal University, China). *cesa1* (also known as *rsw1-1*) (C1646T) was obtained from Yang Zhao (CAS Center for Excellence in Molecular Plant Sciences). The binary constructs *pARH1::gARH1-YFP*, *pFEI1::gFEI1-YFP*, and *pFEI2::gFEI2-YFP* were generated by introducing the corresponding genomic sequence into one of our modified gateway constructs, *pBIB-BASTA-GWR-YFP*[38]. Information regarding the primers used for genotyping and DNA cloning can be found in the supplementary information (Supplementary Tables 1 and 2).

For seedling preparation, surface-sterilized Arabidopsis seeds were maintained in a 4 °C refrigerator for two days for stratification treatment and geminated vertically on half-strength Murashige and Skoog (1/2 MS) agar (1% w/v) plates supplemented with 1% sucrose (w/v) at 22 °C in a 16-h light/8-h dark growth chamber.

### Root hydrotropism treatment
To prepare a split-agar medium with a moisture gradient for hydrostimulation analysis, 1/2 MS medium supplemented with 1% sucrose and 1% agar (w/v) was added to a Petri dish. After solidification, the right bottom half of the medium was removed and replaced with 1/2 MS medium supplemented with 1% sucrose, 1% agar (w/v), and various concentrations of D-sorbitol. Hydrotropism assays were performed as shown in Supplementary Fig. 1 using four-day-old plants in a split-agar system modified from a previous report (Supplementary Fig. 1a–c)[58].

### Root orientation determination during imaging
It is important to maintain the orientation of the roots during imaging. After hydrostimulation, the orientation of most of the roots can be distinguished under a dissecting microscope[59]. Usually, the side of the root with a lower water potential shows an obvious convex phenotype compared with that with a higher water potential. After the plants were transferred to slides with or without staining, the original orientation could be easily retrieved based on the morphology of the roots. For confocal analysis, the roots can be hydrostimulated on slides, and life images can be taken without moving the seedlings. The split-agar medium was added to concave slides, four-day-old plants were moved to medium similar to that used for the split-agar in Petri dishes, and the root tips were hydrostimulated in an airtight box before imaging.

### Measurements of the cortex cell numbers in the root meristematic zone and fluorescence intensity
To determine the number of cortex cells in the meristematic zone, we imaged the root tips stained with propidium iodide via a confocal laser scanning microscope. The fluorescence intensity was measured using ImageJ quantification tools (Supplementary Fig. 1d–g)[59]. Within 200 μm above the quiescent center, each of the fluorescence peaks in the cortex indicates a cell wall.

### Osmotic stress treatment
Four-day-old plants were transferred from 1/2 MS medium to 1/2 MS medium supplemented with D-sorbitol and incubated for 2 h. Root tips stained with propidium iodide were imaged by a confocal laser scanning microscope. The dead cell areas were measured using ImageJ software.

### Cell wall biochemical analyses
The biochemical analyses of cell wall polysaccharides were performed as previously described[60]. Approximately 5-mm-long root tips from four-day-old plants grown on Petri plates were collected. The root tips were rapidly frozen in liquid nitrogen and ground into powder. After the addition of 1.5 mL of 70% ethanol, the mixture was vortexed thoroughly and centrifuged at 9600 × $g$ for 10 min to precipitate the alcohol-insoluble residue. The supernatant was discarded, and 1.5 mL of chloroform/methanol (1:1 v/v) solution was added to the residue. The tube was shaken thoroughly to resuspend the pellet, the mixture was centrifuged at 9600 × $g$ for 10 min, and the supernatant was discarded. The pellet was resuspended in 500 μL of acetone and air dried at 35 °C. The sample was resuspended in 1.5 mL of 0.1 M sodium acetate buffer (pH 5.0) and incubated for 20 min at 80 °C in a heating block. The suspension was cooled on ice, and 35 μL of 0.01% sodium azide ($NaN_3$), 35 μL of amylase (30 U/mg; from *Aspergillus oryzae*, SIGMA, 10065), and 17 μL of pullulanase were sequentially added

(SIGMA, E2412). The tube was closed and vortexed thoroughly. The suspended solution was incubated overnight at 37 °C in a shaker and heated at 100 °C for 10 min in a heating block to terminate the digestion. The mixture was centrifuged (9600 × *g*, 10 min), and the supernatant, which contained solubilized starch, was discarded. The remaining pellet was washed three times by adding 1.5 mL of water. The pellet was resuspended in 500 μL of acetone and air dried at 35 °C. The dried product, which represented the isolated cell wall, was weighed and named the alcohol-insoluble residue (AIR).

The crystalline cellulose amount was measured according to a previously published paper[60]. Add 20 μL inositol solution (5 mg/mL) as internal standard and 250 μL acetone to the pellet. After air dried, 250 μL 2 M trifluoroacetic acid (TFA) was added to the pellet and incubated at 121 °C for 90 min. Cool the samples on ice and centrifuge at 9600 × *g* for 10 min. The pellet can then be used for the crystalline cellulose assay. Add 1 mL Updegraff reagent (Acetic acid: nitric acid: water, 8:1:2 v/v) to the pellet and heat at 100 °C for 30 min. Centrifuge at 9600 × *g* for 15 min, discard supernatant. Wash the pellet with water once and acetone for three times. Air dry the pellet gently. Add 175 μL 72% sulfuric acid and incubate at room temperature for 45 min. Add 825 μL water and then centrifuge at 9600 × *g* for 5 min. The glucose content of the supernatant is quantified using the colorimetric anthrone assay. Add 200 μL of freshly prepared Anthrone Reagent (Anthrone dissolved in concentrated sulfuric acid, 2 mg anthrone/mL sulphuric acid) and incubate at 80 °C for 30 min. Measure the absorbance of samples at a wavelength of 625 nm. The glucose concentration (and consequently the crystalline cellulose content) is calculated based on the absorbance relative to the standard curve. The matrix polysaccharide composition was measured by Biotree Biotech Co., Ltd (http://www.biotree.cn). Uronic acids were calorimetrically measured using 2-hydroxydiphenyl as a reagent and galacturonic acid as the standard[61,62]. The degree of pectin methyl esterification was measured by detecting the methyl esters released from methyl-esterified pectin[63–65].

### Ruthenium red staining

To visualize the seed coat mucilage, dried Arabidopsis seeds were immersed in deionized water, shook at 16 °C for 2 h, and stained with 0.01% (m/v) ruthenium red for 10 min. After staining, the seeds were washed three times with distilled water. The stained seeds were photographed under a microscope with a bright field.

Four-day-old plants were stained with 0.05% (m/v) ruthenium red for 15 min. After staining, the plants were washed three times with distilled water. Subsequently, the stained root tips were imaged by a microscope with a bright field.

### Fluorol yellow staining

The seedlings were incubated in Fluorol Yellow 088 (0.01% in methanol) for 3 days for root tip cutin staining[31]. The seedlings were then counterstained with aniline blue (0.5% in water) for at least 1 h at room temperature and rinsed three times with distilled water. The root tips were photographed by a confocal laser scanning microscope using a laser wavelength of 488 nm.

### Transmission electron microscopy

Two-day-old root tips were fixed in a 2.5% glutaraldehyde solution in 0.1 M phosphate buffer, pH 7.4 (PB buffer), overnight at room temperature and washed with PB buffer (3 × 15 min). The samples were embedded in 1% osmium tetroxide for 2 h at room temperature and washed with PB buffer (3 × 15 min). Gradual dehydration steps were then performed in a series of ethanol solutions (30%, 50%, 70%, 80%, 90%, 100% and another 100% ethanol) for 15 min each. The samples were soaked in acetone (2 × 30 min) and polymerized with acetone/epoxy resin (3 v/1 v) for 1 h, acetone/epoxy resin (1 v/1 v) for 3 h, and epoxy resin overnight. The root tips were ultrathinly sectioned transversely to

a thickness of 50 nm using a Leica slicer (Leica EM UC7) with diamond knives (DiATOME, Switzerland). The slices were stained with uranium acetate and lead citrate for 3 min each. The cell wall, cutin, and wax of the root tips were visualized under an FEI Talos F200C transmission electron microscope (FEI, Talos F200C, Czech Republic).

### Wax extraction and GC–MS analysis

The root cap wax was extracted from 5-mm-long root tips of four-day-old Col-0 plants and from the various mutant and overexpression plants. The root tips were subsequently transferred to glass bottles and immersed in chloroform for 30 s. The liquid was transferred to a new glass bottle. The residue was dried and weighed. After addition of an internal standard, *n*-tetracosane, the solvent was dried using nitrogen gas. The wax was dissolved in a solution mixture of bis-*N,N*-trimethylsilyl trifluoroacetamide (BSTFA, Sigma) and pyridine (1:1, v/v) and then incubated at 100 °C for 30 min. The solution mixture was then dried using nitrogen gas and dissolved in *n*-hexane. The composition and amount of wax were measured by GC–MS (Thermo Fisher trace 1300, USA) with a 30-m HP-5MS column (0.25 mm × 0.25 mm). The GC analysis was performed with the following program: helium carrier gas flow rate of 1.5 mL/min, injection at 260 °C for 10 min, and the temperature was increased at 2 °C/min to 300 °C for 5 min and at 5 °C/min to 320 °C and maintained at 320 °C for 2 min. A C7-C40 saturated alkane mixture was used as a standard reference for alkanes (Sigma 49452-U).

### Cutin extraction and GC–MS analysis

Root tip cutin was extracted from 5-mm-long root tips of two-day-old Col-0, various mutants, and transgenic plants. The root tips were collected in glass tubes, and the extraction method was adapted from Berhin et al.[31]. Briefly, samples were initially washed with isopropanol/0.01% butylated hydroxytoluene (BHT). Sample delipidization was performed three times (1, 16, 8 h) in each of the following solvents: chloroform-methanol (2:1), chloroform-methanol (1:1), and methanol with 0.01% BHT. Samples were then dried for 3 days under vacuum. Add 2 mL reaction medium (20 mL reaction medium is composed of 3 mL methyl acetate, 5 mL of 25% sodium methoxide in methanol, and 12 mL methanol). Internal standards of 5 μg methyl heptadecanoate and 10 μg ω-pentadecalactone/sample were added into the samples. After incubated at 60 °C for 2 h, 3.5 mL dichloromethane, 0.7 mL glacial acetic acid, and 1 mL 0.9% NaCl (w/v) Tris 100 mM pH 8.0 were added. After centrifugation (1500 × *g* for 2 min), the organic phase was collected and dried under a stream of nitrogen. The cutin monomer fraction was derivatized with BFTSA/pyridine (1:1) at 70 °C for 1 h. The solution mixture was then dried using nitrogen gas and dissolved in *n*-hexane. The samples were injected into a 30-m HP-5MS column (0.25 mm × 0.25 mm) on a gas chromatograph coupled to a mass spectrometer (Thermo Fisher Trace 1300, USA). The temperature cycle of the oven was set as follows: 2 min at 50 °C, increment of 20 °C/min to 160 °C, increment of 2 °C/min to 250 °C, and increment of 10 °C/min to 310 °C, held for 15 min.

### Lipidomic analysis

Approximately 5-mm-long root tips were collected from four-day-old Col-0 and *arh1-2 fei1-C fei2-C* triple mutants and weighed. Lipidomic analysis was conducted by BGI Biotech Co., Ltd (https://www.bgi.com). Lipid extraction was performed as follows. The root tips were rapidly frozen in liquid nitrogen. Subsequently, 800 μL of a mixture of methanol (MeOH) and methyl tert-butyl ether (MTBE) at a ratio of 3:1 was added to each 100-mg sample. Additionally, 10 μL of an internal standard was added. The samples were ground using a tissue grinder at 50 Hz for 5 min, ultrasonicated in a water bath for 15 min, and precipitated in a −20 °C refrigerator for 2 h. The samples were treated with 0.5 mL of extractant solution (H$_2$O/MeOH = 3:1). The mixture was vortexed for 1 min. The samples were then centrifuged at 4 °C and

25,000 × *g* for 10 min. Six hundred microliters of the supernatants was collected and dried. To reconstitute the lipids, 400 μL of a lipid reconstitution solution was added to each dried sample, and each mixture was then sonicated in an ice bath for 5 min and centrifuged at 4 °C and 25,000 × *g* for 10 min. The supernatant was then collected.

A Waters UPLC I-Class Plus (Waters, USA) tandem Q Exactive high-resolution mass spectrometer (Thermo Fisher Scientific, USA) was used to separate and detect lipids. Chromatographic separation was performed on a CSH C18 column (1.7 μm 2.1 × 100 mm, Waters, USA). In the positive-ion mode, mobile phase A consisted of 60% acetonitrile in water and 10 mM ammonium formate, and mobile phase B consisted of 90% isopropanol, 10% acetonitrile, 10 mM ammonium formate, and 0.1% formic acid. In the negative ion mode, mobile phase A consisted of 60% acetonitrile in water and 10 mM ammonium formate, and mobile phase B consisted of 90% isopropanol, 10% acetonitrile, and 10 mM ammonium formate. The column temperature was maintained at 55 °C. The gradient conditions were as follows: 40-43% B over 0-2 min, 43-50% B over 2-2.1 min, 50-54% B over 2.1-7 min, 54-70% B over 7-7.1 min, 70-99% B over 7.1-13 min, 99-40% B over 13-13.1 min, maintained at 99-40% B over 13.1-15 min and washed with 40% B over 13.1-15 min. The flow rate was 0.4 mL/min, and the injection volume was 5 μL. Using Q Exactive (Thermo Fisher Scientific, USA), primary and secondary mass spectrometry data were acquired. The full scan range was 70-1050 m/z with a resolution of 70,000, and the automatic gain control (AGC) target for MS acquisitions was set to 3e6 with a maximum ion injection time of 100 ms. The top 3 precursors were selected for subsequent MSMS fragmentation with a maximum ion injection time of 50 ms and resolution of 17,500, and the AGC was 1e5. The stepped normalized collision energy was set to 15, 30 and 45 eV. The ESI parameters were set as follows: sheath gas flow rate, 40×; auxiliary gas flow rate, 10; positive-ion mode spray voltage (|KV|), 3.80; negative-ion mode spray voltage (|KV|), 3.20; capillary temperature, 320 °C; and auxiliary gas heater temperature, 350 °C.

## Toluidine blue staining
Four-day-old plants were immersed in 0.005% (w/v) toluidine blue solution for 1 min and then washed with water for 1 min. The root tips were imaged within 5 min. The staining intensity was measured using ImageJ[59].

## RT–qPCR
Approximately 5-mm-long root tips were collected from 4-day-old plants for total RNA extraction and RT-qPCR analysis. The primers used for RT-qPCR are listed in the supporting information (Supplementary Table 3).

## Pectinase treatments
Three-day-old plants were transferred from 1/2 MS medium supplemented with 1% agar (w/v) and 1% sucrose (w/v) to the same medium supplemented with 50 μM pectinase (pectinase from *Aspergillus aculeatus*, Sigma, P2611) and pretreated for 1 day (as a control, the same volume of DMSO was added as the solvent for pectinase). The seedlings were transferred to hydrostimulating media supplemented with ᴅ-sorbitol on the bottom right side and 50 μM pectinase on both sides and cultured for an additional day. For osmotic stress treatment, pectinase-pretreated plants were transferred to the same medium supplemented with 0 mM or 200 mM ᴅ-sorbitol, treated for an additional 2 h, and subjected to propidium iodide staining. The samples were photographed by a confocal laser scanning microscope.

## Statistical analysis
The data were analyzed using Student's *t* test or one-way ANOVA. Student's *t* test, by two-side without making any adjustments for multiple comparisons, was used to determine whether two datasets were significantly different. "***" indicates a significant difference, and "ns" indicates no significant difference between two datasets based on Student's *t* test at $P < 0.001$. ANOVA was used if more than two datasets were being compared. One-way ANOVA with Tukey's multiple comparison test was applied to each dataset based on a *P* value of 0.001. Letters were used to indicate the significance levels between all the datasets. The same letter indicates no significant difference. However, different letters indicate significant differences between the datasets. Figures with boxplots, boxplots span the first to the third quartiles of the data, and whiskers indicate the minimum and maximum values. The line in the box represents the mean. The exact *P*-value was included in the Source Data file.

## Reporting summary
Further information on research design is available in the Nature Portfolio Reporting Summary linked to this article.

## Data availability
The authors declare that all data supporting the findings of this study can be found within the paper and its Supplementary Files. Source data are provided with this paper. Additional data supporting the findings of this study are available from the corresponding author (J.L.) upon request. Source data are provided with this paper.

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

## Acknowledgements

We thank the Arabidopsis Biological Resource Center (ABRC) for providing the T-DNA insertion mutants of *arh1-1*, *arh1-2*, *fei1*, *fei2*, and *bdg* used in this study. We are grateful to Zhongnan Yang (Shanghai Normal University, China) for providing *kcs7 kcs15 kcs20 kcs21* quadruple mutants and Yang Zhao (CAS Center for Excellence in Molecular Plant Sciences) for providing *cesa1*. Our appreciation also goes to Liang Peng, Yahu Gao, and Li Xie (Core Facilities in the School of Life Sciences, Lanzhou University), for their excellent technique assistance. We thank Huiping Yao, Ning Zhang, and Hao An (Electron Microscopy Centre, Lanzhou University) for their technical help. This study was supported by National Natural Science Foundation of China grants 32030005 to J.L. and 32100261 to J.C.

## Author contributions

J.L. supervised the entire project. J.C. and J.L. designed the experiments. J.C. carried out most of the experiments. X.L., J.S., J.H., L.W. and X. Z. performed part of the experiments. J.L. and J.C. wrote the manuscript.

## Competing interests

The authors declare no competing interests.
