## [Peer Review File · Nature Communications]

Defects in the cell wall and its depositions caused by loss-of-function of three RLKs alter root hydrotropism in *Arabidopsis thaliana*Reviewer #1 (Remarks to the Author):

In this study the authors screened overexpressors of LRR-RLKs and found three of them had less sensitivity to osmotic pressure gradient, while they had tolerance to osmotic stress. Coincided with this observation, their analyses revealed that the corresponding triple mutant displayed an enhanced response to the water potential gradient and less tolerance to osmotic stress. In this triple mutant, cell wall, wax and cuticle production were defective, which was further confirmed by their biochemical and transcriptomic analyses. Because other mutants defective in biosynthesis of cell wall, wax, and cuticle displayed phenotypes similar to the triple mutant, they concluded that an adequate cell wall, wax and cuticle synthesis plays important roles in regulating hydrotropism. The topic is truly interesting and their massive analyses from various aspects are indeed valuable. However, I feel that their conclusion is not fully supported because of the following reasons.

1. The cause-and-effect relationship between cell wall integrity and regulation of hydrotropic sensitivity is not established. Similarly, I understood that excess wax or cuticle production observed in overexpressors of LRR-RLKs can enhance osmotic stress tolerance, however it does not directly mean that such overproduction of the materials regulates the osmotic tolerance. In other words, one can also explain the authors' results as a "side effect" of malfunctional cell wall. The term "regulate" should be used more carefully.
2. Related to the comment 1, root gravitropism of the mutants should be examined, for the rate of gravitropic response can alter the degree of hydrotropic root curvature. Considering this fact, the alterations of root hydrotropic curvature can simply be explained by the alterations of root gravitropism caused by cell death which is induced by osmotic stress.
3. In the previous report, the authors demonstrated that asymmetric division of root meristematic cells are important in root hydrotropism. In addition, a report by Dietrich et al. had shown that root cortical cells at elongation zone are responsible for hydrotropism. On the other hand, it seems that in the triple mutant, these cells are very susceptible to osmotic stress. I cannot understand how each root could regulate its hydrotropism when these cells are severely damaged.

Other minor comments.

1. As far as I know, the mechanism of root hydrotropism differs among the plant species. The authors should describe the fact carefully.
2. In each experiment, the number of biological replicates should be described.
3. Data of time-course should be presented.

Reviewer #2 (Remarks to the Author):

The manuscript by Chang et al describes an experimental approach to identify genes involved in hydrotropism. They identified three RLKs that reduced hydrotropism when overexpressed and mutations in these genes increase hydrotropism. Two of them, previously shown to influence cellulose production, FEI1 and FEI2, are located at the plasma membrane facing the soil. Most importantly, they show that mutations in the three genes strongly reduce wax and cutin production, lipid content, and alter cell wall composition.

Gene expression and metabolome analyses revealed that indeed lipid metabolism and some cell wall-related genes are altered, which is particularly consistent for the lipid metabolism, whereas the data shown for the gene expression analysis is limited and might represent a very selective subfraction that fits the general picture the authors want to bring across.

The manuscript contains interesting data that will be worth publishing. At the other hand, there are a number of points that need to be addressed.

Major points

Throughout the manuscript, the authors often write that the RLKs affect cell wall modification and cutin/wax. The term cell wall modification includes way more things than just cutin and wax deposition. This way of description is misleading. For example, on line 325 ff, they say that cell wall modifications are observed in areal parts of the plants. As if cutin/wax deposition would be the only existing cell wall modification. Any primary cell wall is modified, at any time! Better refer to deposition of cutin/wax onto the cell wall. And cell wall composition should be treated separately. Also, they seem to assume that the RLKs indeed cause all these changes and that the RLKs are involved in cell wall integrity sensing (e.g. line 246). I consider it as much more likely that

cutin/wax production is affected and the changes in the cell wall are secondary effects. Perhaps the gene expression analysis would give a better insight. But since there are only few genes shown (those that fit the hypothesis..?), I can't tell.

It would be helpful to have qRT-PCR data on the different mutants. It is surprising that the T-DNA mutants cannot be obtained as triple mutant, but the crispr mutants can. While I don't doubt the finding, there should be some more information on this point.

In suppl. Fig 6, a triple mutant root tip is shown – it is much enlarged, not to say gigantic compared to the wild type. Most interestingly, this major change is not observed in any other picture of the triple mutant. how come? Perhaps I don't understand the term hydrostimulation? But the triple mutant remains in the normal MS, hence under the same conditions, yet is gigantic. From suppl. Fig. 4, it seems that the arh1 single mutant has comparable effects as the double and triple mutants. It would be interesting to see the data of the fei1 fei2 double mutant. Also, it is surprising to see that the triple mutant can be complemented with all three genes under their own promoter – considering the tremendous differences in expression patterns. Please explain.

In Fig. 2, they show that ARH1 accumulates in newly forming cell plates during cell division within the root. This does not really go well together with playing a role in cutin/wax deposition on the root epidermis. Please explain.

In general, there is no description of growth phenotypes of the triple mutant or the overexpression lines. Considering the much reduced lipid content of these plants, how can they survive? Or is it lipid deposits that are affected rather than the plasma membrane content?

It is not always clear what the authors mean with "root tip area"; root cap? Tip area? Including elongation zone? How do they explain their findings and those of Ref. nr. 30, where it is explicitly stated that cutin/wax is found in the root cap but not elsewhere in the root?

While it is written in a generally understandable way, I think that the manuscript would benefit from a professional English editing to bring the language to the same level as the science presented.

The discussion does not address the points that don't go well together but rather turn around Darwin and his studies with plants – which is historically interesting but does not provide a better insight into the findings described in the paper. They also seem very convinced that the three RLKs really directly have the described effects on all the processes (e.g. lane 246). A less biased view and alternative explanations would be useful.

The title should be changed such that the effect of the RLKs on cutin/wax formation and FA metabolism is stressed. It appears that this is the primary effect – at least based on the data provided.

Minor points.

Lane 36-40 Auxin should not be involved in hydrotropic growth. But then, the pin2 mutant (which changes auxin distribution) is affected in hydrotropic growth. That doesn't stick.

Lane 41-45. What is the activity MIZ1? That and the biological relevance to the topic is not clear. There's a mix up of supplementary figures. In Fig. S2, a phylogenetic tree is shown that is not corrected mentioned in the text. In any case, I doubt that this is an extremely helpful figure given that it is not a point in the text.

Lane 244. Expansins and Extensins are very different proteins with very different activities!!

Suppl. Fig. 7: The section on cytokinins can be removed. It opens yet another avenue but is not being followed up in any way. As is, it is very insulated and does not provide much insight.

Reviewer #3 (Remarks to the Author):

Summary

The manuscript by Chang et al. presents findings on the function of three RLKs in root hydrotropism through modulation of root tip cell walls and cuticle as well as asymmetric distribution of cytokinins during hydrotropism responses. Gain and loss of function lines were employed for phenotypic measures of root hydrotropism responses. A triple knockout, arh1-2fei1-Cfei2-C, that was generated through CRISPR-Cas9 editing of FEI1 and FEI2 in the arh1-2 background was used for the majority of biochemical, microscopic, and molecular-genetic characterizations. The arh1-2fei1-Cfei2-C triple mutant was found to have hypersensitive hydrotropic responses whereas as individual overexpression lines exhibited a lack of or reduced

hydrotropic responses. Further analyses showed a reduced deposition of cell wall and cuticle in the *arh1-2fei1-Cfei2-C* triple mutant as well as reduced transcript abundance of cell wall, lipid, and cuticle biosynthesis gene transcript abundance. Lastly, analysis of cell wall and cuticle-defective mutants and pectinase-treated WT plants revealed elevated hydrotropic responses and increased sensitivity to osmotic stress.

Review

The authors have presented an interesting collection of phenotypic, biochemical, and molecular-genetic characterization of these three RLKs, especially the *arh1-2fei1-Cfei2-C* triple mutant. The observed phenotypes, especially the biochemical phenotypes, are quite exceptional. I believe that the findings present herein have potential to improve our understanding of the relationship between cell wall modifications and root physiological responses. Furthermore, the results point to a potentially novel discovery on the regulation of cell wall through RLK-mediated signaling.

With that said, I do have some major concerns that put into question some of the conclusions that are presented in the manuscript.

First, the RNA-seq data merit further analyses and presentation of those analyses. It is important for the reader to be able to observe and assess all differentially expressed genes (DEGs) from the RNA-seq data set, as supplemental data, instead of only select genes of interest. This is particularly important given that the primary genes under investigation are Receptor Like Kinases that have the potential to regulate multiple biosynthetic and signaling pathways. I believe that this is required for the reader to be able to fully assess the conclusions drawn in the manuscript. It also isn't clear if the RNA-seq data have been deposited to NCBI's Sequence Read Archives (SRA) or a similar repository. This is generally a standard practice for manuscripts that present RNA-seq data.

The presented lipidomics data are particularly concerning. No description of the lipid extraction methods is provided. This is particularly relevant for the presented classes and molecular species of lipids presented in the manuscript (Figure S18). Phosphatidylcholine (PC) should be the dominant class of lipids in roots (and other Arabidopsis tissues). In roots, phosphatidylcholine typically constitutes ~45 mole % of total root lipids (Li-Beisson et al., 2013, *Acyl Lipid Metabolism, The Arabidopsis Book*). No PC is presented in the lipidomics data (Figure S18). Furthermore, there appears to be a large preponderance of Phosphatidic Acids (PA), Lyso-lipids (lyso-phosphatidic acid, lyso-phosphatidylethanolamine, lyso-phosphatidylmethanol, lyso-phosphatidylethanol), phosphatidylmethanol and phosphatidylethanol. These are clear indications of improper extraction of the lipids. It is critical to inactivate the phospholipases by quenching the tissues with hot iso-propanol. In roots, there should be close 0% PA (Li-Beisson et al., 2013, *Acyl Lipid Metabolism, The Arabidopsis Book*). Furthermore, the presence of lyso-phosphatidylmethanol is another clear indication of improper lipid extraction. "...phosphatidylmethanol which can be formed by the action of endogenous phospholipase D when plant tissue is extracted with methanolic solvents at or above room temperature (Roughan et al., 1978)" (Miquel & Browse, 1992, *JBC*, Vol. 267, No. 3, pp. 1502-1509). As such, no conclusions can be drawn from the lipidomics data.

Lines 206 – 209: The authors state, "Surprisingly, we found a layer of electron-transparent cell wall modification covering the cuticle in the root caps of Col-0. We assumed that it is wax. This assumption was confirmed by using a *kcs7 kcs15 kcs20 kcs21* quadruple mutant as a negative control, in which the wax biosynthesis pathway is blocked and such a layer was no longer observed in the quadruple mutant (Fig. 3l, p, and t)".

The electron transparent layer that is being referred to (Figure 3m for WT) looks more like a fracture in the epoxy resin than a true anatomical feature of the plant. It is quite common for epoxy resin to fracture near hydrophobic surfaces. To support this conclusion, the authors would need to present multiple images from multiple, independent biological samples.

Furthermore, the authors may want to revise their understanding of the basic anatomy of plant cuticles. Plant cuticles are comprised of a cutin polymer that is embedded and covered with waxes. The cuticle is subdivided into different layers including the cuticular layer which represents the transition between a carbohydrate-rich cell wall and lipidic cutin polymer, the cuticle proper which is comprised largely of cutin that is embedded with intracuticular waxes, and an epicuticular wax

layer (Yeats & Rose, 2013, *Plant Physiology*, 163, pp. 5–20). It is very difficult to ascertain what is “wax” from TEM images. Epicuticular waxes can be visualized with SEM.

Line 342: “On one hand, thicker cell wall, cuticle, and wax...” Wax is part of the cuticle. The cuticle is comprised of a cutin polymer that is embedded and covered with cuticular waxes.

To the best of my knowledge, a cuticular wax phenotype for the *kcs7 kcs15 kcs20 kcs21* quadruple mutant has never been verified (not in leaves, stems, or, relevant to this manuscript, root cap cuticles). The paper cited for this mutant refers to a *kcs7 kcs15 kcs21* triple mutant that has defects in pollen tapetal lipid deposition and has not been characterized for other cuticular phenotypes (i.e. root cap cuticle) (Zhang et al., 2021, *Front. Plant Sci., Sec. Plant Physiology*, Volume 12, <https://doi.org/10.3389/fpls.2021.770311>). I think that it would behoove the authors to provide evidence and characterization of the cuticular defects of the *kcs7 kcs15 kcs20 kcs21* quadruple mutant.

This also highlights another important consideration of the manuscript. The cuticle work is primarily focused on cuticular wax. However, the presented RT-qPCR illustrates downregulation of several cutin-related biosynthesis genes yet no chemical characterization of the root cap cutin of the *arh1-2fei1-Cfei2-C* triple mutant or *kcs7 kcs15 kcs20 kcs21* quadruple mutant is presented. Again, the authors have not clearly delineated the difference between cutin and wax which may be why these data are not presented. Given that cuticle is a major focus of the manuscript, it would perhaps be best to present some characterization of the cutin composition of the *arh1-2fei1-Cfei2-C* triple mutant and *kcs7 kcs15 kcs20 kcs21* quadruple mutants.

Lines 319 – 321: the authors refer to a potential function for ARH1-mediated signaling to control the expression of “yet to be identified” transcription factors. I encourage the authors to revisit the RNA-seq data and look for transcription factors known to regulate cuticle, cell wall, etc. Inclusion of a CSV or Excel file with the DEGs would be helpful.

Lines 332 – 337: The authors state, “Interestingly, we also identified a layer of electron-transparent cell wall modification laying outside of the cuticle which was only identified in the root cap region but not in the elongation zone (Fig. 3i-t). We then confirmed this identification by mutant analysis and composition determination by GC-MS that this layer of modification is actually wax.” This isn’t entirely accurate. Please see my comments about the TEM above. The transparent layer looks like a fracture in the epoxy resin and it is difficult to ascertain what is a “wax” layer from TEM. To verify this, many sections from multiple biological replicates would be required. If this transparent layer truly is “wax”, TEM analysis before and after wax extraction would be required to validate this conclusion.

Subcellular localization experiments, Figure 2: Although it is expected that RLKs would localize to the PM, the presented confocal images aren’t sufficient to demonstrate PM localization. A PM marker should be used to validate co-localization.

Subcellular localization experiments, Figure S13, Lines 178-179: “The distribution and localization of ARH1-YFP, FEI1-YFP, and FEI2-YFP were not altered by hydrostimulation treatment (Supplementary Fig. 13).” It is very difficult to ascertain this from the presented images. Based on the images presented in Figure S13, hydrostimulation of *pFEI2::gFEI2-YFP* plants appears to stimulate promoter activity in cortical cells and maybe even vasculature. Clearer images would be helpful.

Methods, Fluorol yellow staining: What emission wavelength or filter was used for confocal imaging?

Minor concerns

Figure 1n, Lines 145 – 147. “Root tips of *arh1-2 fei1-C fei2-C* triple mutant were swelled after seedlings were transferred from normal 1/2 MS medium to a split 1/2 MS medium supplemented with 800 mM D-sorbitol...” The manuscript text says 800 mM, but the figure is labeled as 400 mM. Please clarify. Figure 1n is also lacking the red arrows that delimit the root zone used for cell counts.

Line 268: KCS enzymes are referred to as "wax synthetases". KCS enzymes are beta-ketoacyl CoA synthases.

Conclusions

Chang et al. have presented an interesting collection of phenotypes that potentially points to a role for ARH1 in regulating root hydrotropism through modulating cell wall and cuticle. However, there are some major concerns that should be addressed to provide better support for the conclusions stated in the manuscript. I recommend the following to the authors: 1) to provide a more comprehensive analysis of the RNA-seq data including a CSV or Excel file with all DEGs, 2) to clearly differentiate between wax, cutin, and cuticle, 3) to employ a series of mutants defective specifically and uniquely in cutin, wax, specific cell wall carbohydrate domains, membrane lipids, etc. to better understand the relative contribution of each to root hydrotropic responses, 4) to analyze the cutin content of arh1-2 fei1-C fei2-C triple mutant, 5) to characterize the kcs quadruple mutant in terms of wax and cutin content and composition (and employ mutants specifically affected in cutin versus wax), 6) to provide better support for PM localization of ARH1, FEI1, and FEI2 through the use of PM markers, and 7) repeat the lipidomics with proper extraction conditions (or utilize simpler lipid analyses with proper extraction conditions). Further analysis of the RNA-seq data will be revelatory and may present opportunities for a better and more comprehensive investigation of the function of ARH1, FEI1, and FEI2 through further experimentation. This may alter the trajectory of the work and will likely require further discussion.

Responses to reviewers

Here list our detailed responses to all the questions from three anonymous reviewers point-by-point. The figures presented in this reponse are newly obtained data. Because some questions from different reviewers are similar, the answers and corresponding figures may show up repetitively. All changes in our revised manuscript are highlighted in yellow.

Reviewer #1 (Remarks to the Author):

In this study the authors screened overexpressors of LRR-RLKs and found three of them had less sensitivity to osmotic pressure gradient, while they had tolerance to osmotic stress. Coincided with this observation, their analyses revealed that the corresponding triple mutant displayed an enhanced response to the water potential gradient and less tolerance to osmotic stress. In this triple mutant, cell wall, wax and cuticle production were defective, which was further confirmed by their biochemical and transcriptomic analyses. Because other mutants defective in biosynthesis of cell wall, wax, and cuticle displayed phenotypes similar to the triple mutant, they concluded that an adequate cell wall, wax and cuticle synthesis plays important roles in regulating hydrotropism.

The topic is truly interesting and their massive analyses from various aspects are indeed valuable. However, I feel that their conclusion is not fully supported because of the following reasons.

1. The cause-and-effect relationship between cell wall integrity and regulation of hydrotropic sensitivity is not established. Similarly, I understood that excess wax or cuticle production observed in overexpressors of LRR-RLKs can enhance osmotic stress tolerance, however it does not directly mean that such overproduction of the materials regulates the osmotic tolerance. In other words, one can also explain the authors' results as a "side effect" of malfunctional cell wall. The term "regulate" should be used more carefully.

Our Response: Thanks for the suggestion. The original title may be a little bit misleading. Therefore, we modified the title in our revised version which may explain the cause and effect more clearly. What we found in this manuscript is actually that the three LRR-RLKs directly regulate the biosynthesis of cellulose, pectin, cuticle, and wax at root tips. The triple mutant showed defects in the cell wall and its depositions. The altered hydrotropic response and osmotic response are the indirect effects. This view is supported by the fact that all the defective mutants of cellulose, cuticle, and wax also display altered hydrotropic and osmotic responses, similar to the triple mutant of *LRR-RLKs*. The relationship of cause and effect has been clarified in the revised manuscript (Page 14, line 257-258, and Page 15, line 290-294).

2. Related to the comment 1, root gravitropism of the mutants should be examined, for the rate of gravitropic response can alter the degree of hydrotropic root curvature. Considering this fact, the alterations of root hydrotropic curvature can simply be explained by the alterations of root gravitropism caused by cell death which is induced by osmotic stress.

Our Response: Thanks for the question. We carried out two different gravitropic response experiments. In the first experiment, we compared the root gravitropic response of four-day-old wild type Col-0 and the triple mutant, *arh1-2 fei1-C fei2-C*, on normal 1/2 MS medium (Supplementary Fig. 6). We did not see obvious root gravitropic response differences between the triple mutant and wild type. The new growth of the roots from two genotypes is also similar after the roots were transferred. In the second experiment, we did root gravitropic response on the osmotic medium (1/2 MS supplemented with different concentrations of D-sorbitol). Again, we did not see obvious differences in the roots of the two genotypes (Supplementary Fig. 7). These observations suggest the increased hydrotropic response of the triple mutant is unlikely caused by the altered root gravitropism.

Supplementary Fig. 6 | The triple mutant roots exhibit an enhanced hydrotropic response while maintaining normal gravitropic response and growth rate compared to Col-0.

a-b, Time course analyses of root hydrotropic responses and growth rates of Col-0 and the triple mutant. Four-day-old seedlings were transferred from 1/2 MS medium to hydrostimulating medium containing 400 mM D-sorbitol at the bottom right side of the plates. **c-d,** Time course analyses of gravitropic responses and growth rates of Col-0 and the triple mutant. Four-day-old seedlings were transferred from 1/2 MS medium to a new 1/2 MS medium and horizontally incubated. Four biological replicates were carried out.

Supplementary Fig. 7 | Roots of the triple mutant *arh1-2 fei1-C fei2-C* showed a gravitropic response similar to those of Col-0 under various osmotic stresses.

a, Gravitropic responses of Col-0 and the triple mutant. Four-day-old seedlings of Col-0 and the triple mutant were transferred from 1/2 MS medium to 1/2 MS medium supplemented with various concentrations of D-sorbitol and incubated for 24 hours. **b**, Measurements of root gravitropic responses after a 24-hour gravistimulation treatment under various osmotic stress conditions. Scale bars represent 10 mm. “n” represents the number of roots analyzed in the experiment. Three biological replicates were carried out. One-way ANOVA with Tukey’s multiple comparison test was used for statistical analyses with $P < 0.01$.

3. In the previous report, the authors demonstrated that asymmetric division of root meristematic cells are important in root hydrotropism. In addition, a report by Dietrich et al. had shown that root cortical cells at elongation zone are responsible for hydrotropism. On the other hand, it seems that in the triple mutant, these cells are very susceptible to osmotic stress. I cannot understand how each root could regulate its hydrotropism when these cells are severely damaged.

Our Response: This is a really good question. In this study we found that reduced integrity of cell wall and its depositions (cuticle and wax) in the root tips led to decreased tolerance to osmotic stress but increased hydrotropic response. On the other hand, increasing the biosynthesis of cell wall and its depositions by overexpressing the three *LRR-RLKs* can increase the tolerance of roots to osmotic stress and decrease root hydrotropic response. The cell permeability was also altered. The triple mutant showed increased permeability, while the overexpressors of these three *LRR-RLKs* showed decreased permeability (Supplementary Fig. 27). In the revised manuscript, we added new data to confirm that the overexpressors of these three *LRR-RLKs* did result in the accumulations of cell wall, cuticle, and wax. Our results indicated the new growth of the triple mutant roots on the hydrostimulation plates supplemented by 400 mM D-sorbitol (split-agar plates) is similar to the new growth of wild type (Supplementary Fig. 6). At the meantime, the hydrotropic response of the triple mutant is significantly higher than that of wild type. In the split-agar plates containing 800 mM D-sorbitol, however, severe root tip swellings were found in the triple mutant but not in wild type, suggesting the trade-off between hydrotropic response and osmotic tolerance may only happen under certain range of water potential. If water potential decreased to a certain threshold, the root tip cells of the triple mutant will be killed. Root hydrotropic response and osmotic tolerance will be abolished.

Supplementary Fig. 27 | Roots with defects in cell wall, cuticle, or wax showed enhanced permeability to toluidine blue.

a-h, Representative toluidine blue-stained four-day-old roots of Col-0 and other genotypes. **i**, Staining intensity for the root caps (red box as shown in **c**) of the roots represented in (**a-h**). **j**, Staining intensity for the meristematic region (200 μm starting from the quiescent center, white box as shown in **c**) of the roots represented in (**a-h**). Scale bars represent 50 μm . Each circle represents the measurement of an individual root. “n” represents the number of roots analyzed in the experiment. Three biological replicates were carried out. One-way ANOVA with Tukey’s multiple comparison test was used for statistical analyses with $P < 0.01$.

Other minor comments.

1. As far as I know, the mechanism of root hydrotropism differs among the plant species. The authors should describe the fact carefully.

Our Response: We revised our manuscript and specified that all our experiments were carried out in *Arabidopsis* (Page 3, line 37-42).

2. In each experiment, the number of biological replicates should be described.

Our Response: In our revised manuscript, we added the number of biological replicates for all the experiments we conducted.

3. Data of time-course should be presented.

Our Response: We carried out the time-course experiments for root growth curvature and new growth during root hydrotropic and gravitropic response and new data are included in the revised manuscript (Supplementary Fig. 6, Movie 1, and Movie 2).

Reviewer #2 (Remarks to the Author):

The manuscript by Chang et al describes an experimental approach to identify genes involved in hydrotropism. They identified three RLKs that reduced hydrotropism when overexpressed and mutations in these genes increase hydrotropism. Two of them, previously shown to influence cellulose production, FEI1 and FEI2, are located at the plasma membrane facing the soil. Most importantly, they show that mutations in the three genes strongly reduce wax and cutin production, lipid content, and alter cell wall composition.

Gene expression and metabolome analyses revealed that indeed lipid metabolism and some cell wall-related genes are altered, which is particularly consistent for the lipid metabolism, whereas the data shown for the gene expression analysis is limited and might represent a very selective subfraction that fits the general picture the authors want to bring across.

The manuscript contains interesting data that will be worth publishing. At the other hand, there are a number of points that need to be addressed.

Major points

1. Throughout the manuscript, the authors often write that the RLKs affect cell wall modification and cutin/wax. The term cell wall modification includes way more things than just cutin and wax deposition. This way of description is misleading. For example, on line 325 ff, they say that cell wall modifications are observed in areal parts of the plants. As if cutin/wax deposition would be the only existing cell wall modification. Any primary cell wall is modified, at any time! Better refer to deposition of cutin/wax onto the cell wall. And cell wall composition should be treated separately.

Our response: Thanks for the good suggestions. We have modified the descriptions throughout the entire text and the title.

2. Also, they seem to assume that the RLKs indeed cause all these changes and that the RLKs are involved in cell wall integrity sensing (e.g. lane 246). I consider it as much more likely that cutin/wax production is affected and the changes in the cell wall are secondary effects. Perhaps the gene expression analysis would give a better insight. But since there are only few genes shown (those that fit the hypothesis..?), I can't tell. It would be helpful to have qRT-PCR data on the different mutants. It is surprising that the T-DNA mutants cannot be obtained as triple mutant, but the crispr mutants can. While I don't doubt the finding, there should be some more information on this point.

Our response: We have added the whole RNA-seq data in the source data file. We actually did KEGG analysis and compared the expression profiles of wild type and the triple mutant. We found that the expression levels of the genes in the triple mutant involved in sugar metabolism and fatty acid biosynthesis are most significantly altered in the entire genome. Therefore, we did RT-qPCR for all the genes encoding for the enzymes catalyzing the biosynthesis of cell wall, pectin, cuticle, and wax. It has been clarified in the text (Page 13, line 240-247).

We found the triple mutant could not be obtained if all three genes are null mutants, suggesting that loss of all these three genes can be lethal. Luckily, we obtained a triple mutant designated *arh1-2 fei1-C fei2-C*, in which *arh1-2* and *fei1-C* are the null alleles, but *fei2-C* is a partially loss-of-function mutant. The *fei2-C* is a partially loss-of-function mutant is supported by a few lines of evidence. First, *fei2-C* contains an in-frame deletion in the 5' region of the coding sequence which results in the loss of a 70-aa fragment. Second, we analyzed the expression levels of 21 genes involved the biosynthesis of cellulose, pectin, cutin, and wax in wild type, *fei2-C*, *fei2-C-2* (a weak allele which was obtained recently), and *fei2* (Supplementary Fig. 22). The purpose of this analysis is just to confirm that *fei2-C* and *fei2-C-2* are partially loss-of-function mutants. The selected genes are those showing obvious down-regulation in the *fei2* null mutant. Our analysis confirmed that *fei2-C* and *fei2-C-2* are truly weak mutant alleles of FEI2. Third, we confirmed the obtained defective

phenotypes of the triple mutant is due to the loss-of-function (including partially loss-of-function) of the three genes by complementation analyses.

Supplementary Fig. 22 | Transcriptional levels of genes encoding for enzymes catalyzing the biosynthesis of cell wall, cuticle, and wax in various mutants of *FEI2*.

a-c, The expression levels of the genes in the root tips of *FEI2* mutants. *fei2-C* and *fei2-C-2* were generated through gene editing, and *fei2* is a T-DNA insertion line of *FEI2*. **d**, Gene editing details at the genomic sequence level of *fei2-C-2*. The deleted sequences were marked in gray. **e**, Gene editing details at cDNA level of *fei2-C-2*. Data are the means \pm SD of three biological replicates. One-way ANOVA with Tukey's multiple comparison test was used for statistical analyses with $P < 0.01$.

3. In suppl. Fig 6, a triple mutant root tip is shown – it is much enlarged, not to say gigantic compared to the wild type. Most interestingly, this major change is not observed in any other picture of the triple mutant. how come? Perhaps I don't understand the term hydrostimulation? But the triple mutant remains in the normal MS, hence under the same conditions, yet is gigantic.

Our response: We prepared hydrostimulation medium by replacing the right bottom side of the 1/2 MS with 1/2 MS supplemented with different concentrations of D-sorbitol as shown in supplementary Figure 1. It was reported that D-sorbitol can absorb water, and moisture gradient can be formed along the border of the split media. Higher concentration of D-sorbitol can absorb more moisture and therefore greater water potential difference. We found that 800 mM D-sorbitol at the right bottom side of the medium can induce root tip swellings in the triple mutant but not in wild type (**Supplementary Fig. 9**). Less concentration of D-sorbitol cannot induce root tip swellings in the triple mutant or wild type. We used 800 mM D-sorbitol only in the original Supplementary Figure 6 (it was replaced by Supplementary Figure 9 in the revised manuscript) but not in other figures. We took the images of the root tips of the triple mutant and wild type under a series of concentrations of D-sorbitol split-agar medium and presented in Supplementary Figure 9.

Supplementary Fig. 9 | Root tip cells of the triple mutants exhibited abnormal expansion upon hydrostimulation or osmotic treatment.

a-h, Phenotypes of Col-0 root tips after treated with hydrostimulation or osmotic stress. Four-day-old Col-0 seedlings were transferred from 1/2 MS medium to split 1/2 MS media, supplemented with various concentrations of D-sorbitol at the bottom right side of the medium (a-d), or to osmotic stress media containing different concentrations of D-sorbitol (e-h) and incubated for 24 hours. **i-p**, Phenotypes of the triple mutant root tips after treated with hydrostimulation or osmotic stress. Four-day-old *arh1-2 fei1-C fei2-C* seedlings were transferred from 1/2 MS medium to split 1/2 MS media, supplemented with different concentrations of D-sorbitol at the bottom right side of the media (i-l) or to osmotic stress media containing different

concentrations of D-sorbitol (m-p) and incubated for 24 hours. Three biological replicates were carried out. Scale bars represent 50 μm .

4. From suppl. Fig. 4, it seems that the *arh1* single mutant has comparable effects as the double and triple mutants. It would be interesting to see the data of the *fei1 fei2* double mutant. Also, it is surprising to see that the triple mutant can be complemented with all three genes under their own promoter – considering the tremendous differences in expression patterns. Please explain.

Our response: In Supplementary Figure 4, we only compared the hydrotropic response of wild type, single mutants, and double mutants. In this figure, *arh1-1 fei1 fei2 (+/-)* is not a true homozygous triple mutant because *fei2* is a heterozygous. All the single mutants used in Supplementary Figure 4 were T-DNA inserted mutants. From *arh1-1 fei1 fei2 (+/-)*, we never obtained homozygous triple mutant, likely due to the lethality of the triple mutant. We therefore generated new mutants for *FEI1* and *FEI2* by CRISPR, hoping to get some weak alleles. *fei2-C* is indeed a weak allele of *FEI2* and we eventually obtained homozygous triple mutant *arh1-2 fei1-C fei2-C*. We compared the hydrotropic responses in wild type, single, double, and triple mutants and the data are presented in Supplementary Figure 5. In this figure, we can clearly see the triple mutant has higher hydrotropic response than wild type, single mutants and double mutants.

In the complementation experiments, although we used native promoters of the three *LRR-RLKs*, their expression levels are much higher than those in wild type (Supplementary Fig. 14I). In other words, the transgenes act as overexpressors. We clarified this issue in the revised manuscript.

Supplementary Fig. 14 | Expression of *ARH1*, *FEI1*, or *FEI2* driven by their native promoters can rescue the reduced osmotic tolerance of the triple mutant *arh1-2 fei1-C fei2-C*.

a-j, Representative propidium iodide-stained roots from four-day-old Col-0, *arh1-2 fei1-C fei2-C*, and transgenic seedlings harboring *pARH1::gARH1-YFP*, *pFEI1::gFEI1-YFP*, or *pFEI2::gFEI2-YFP* in *arh1-2 fei1-C fei2-C* background. Seedlings were transferred from 1/2 MS medium to 1/2 MS medium supplemented with 0 mM (a-e) or 200 mM (f-j) D-sorbitol, respectively, and incubated for 2 hours.

k, Measurements of the dead cell areas in a $400 \mu\text{m} \times 100 \mu\text{m}$ region (as shown in figure a) above the quiescent center. **l**, Relative expression levels of *ARH1*, *FEI1*, and *FEI2*, in the triple mutant and complemented transgenic plants. Three biological replicates were carried out. Scale bars represent $50 \mu\text{m}$. Each circle represents the data from an individual root. “n” represents the number of roots analyzed in the experiment. One-way ANOVA with Tukey’s multiple comparison test was used for statistical analyses with $P < 0.01$.

5. In Fig. 2, they show that ARH1 accumulates in newly forming cell plates during cell division within the root. This does not really go well together with playing a role in cutin/wax deposition on the root epidermis. Please explain.

Our response: Although ARH1 is enriched in the newly forming cell plate area, it can also be detected on the plasma membrane of root cap cells. Our chemical analyses in the roots of the *ARH1* overexpressors indicated that ARH1 can induce the depositions of cuticle and wax (Fig. 4, Supplementary Fig. 19). In addition, genes encoding for the enzymes catalyzing the biosynthesis of cutin and wax are also down-regulated in *arh1-1* and *arh1-2* single mutants (Supplementary Fig. 21a, c).

Fig. 4 | Amount of total and each chemical composition of wax and cuticle in the root tips of wild type, overexpressors, and various mutants.

a, Amount of total wax in different genotypes. **b**, Amount of each chemical composition of wax in different genotypes. **c**, **d**, TEM images showing cell wall, cuticle, and wax at the surface of the outermost lateral root cap cells (100 µm from the

root tip) before and after wax extraction. **e**, Amount of total cutin in various genotypes. **f**, Amount of each chemical composition of the cutin in various genotypes. The measurements were carried out by using gas chromatography-mass spectrometer (GC-MS) (a, b, e, f). FA, fatty acids; HFA, ω -hydroxy fatty acids; TriOH 18:1 FA, 9,10,18-triOH C18:1 fatty acids; DCA, dicarboxylic acids. Data are the means \pm SD of three biological replicates (a, b, e, f). Scale bars represent 500 nm in (c, d). One-way ANOVA with Tukey's multiple comparison test was used for statistical analyses with $P < 0.01$.

Supplementary Fig. 19 | Compositions and amount of root cap wax.

a, Carbon-chain-length distributions of alkanes in wax extracted from the 5 mm root tips of Col-0 and other genotypes. **b**, Carbon-chain-length distributions of alcohols, ketones and fatty acids in wax extracted from the 5 mm root tips of Col-0 and other genotypes. Data are the means \pm SD of three biological replicates. One-way ANOVA with Tukey's multiple comparison test was used for statistical analyses with $P < 0.01$.

6. In general, there is no description of growth phenotypes of the triple mutant or the overexpression lines. Considering the much reduced lipid content of these plants, how can they survive? Or is it lipid deposits that are affected rather than the plasma membrane content?

Our response: The triple mutant *arh1-2 fei1-C fei2-C* which we generated in this study showed no obvious growth defects in comparison with wild type under normal growth conditions (Supplementary Fig. 8). The *fei2-C* mutant in this triple mutant is actually a weak allele. We could not obtain a triple mutant with all three LRR-RLKs being completely knocked out, suggesting the essential roles of these three LRR-RLKs for the survival of Arabidopsis plants. We saw the lipids are also altered in the triple mutant but we cannot confirm its contribution to the changes of hydrotropic response because no adequate mutants can be used for the analysis. But we did find the importance of cell wall integrity, and depositions of cuticle and wax at the root tips to root hydrotropic response.

Supplementary Fig. 8 | The triple mutant showed no obvious growth defects compared to wild type under normal growing conditions.

a, Phenotypes of 10-day-old seedlings of Col-0 and *arh1-2 fei1-C fei2-C* grown on 1/2 MS medium. **b**, Root length of 10-day-old Col-0 and *arh1-2 fei1-C fei2-C* seedlings grown on 1/2 MS medium. **c**, Phenotypes of 30-day-old plants of Col-0 and *arh1-2 fei1-C fei2-C* grown in soil. Scale bars represent 10 mm. “n” represents the number of roots analyzed in the experiment. Three biological replicates were carried out. Student’s *t*-test was used for statistical analyses with $P < 0.05$.

7. It is not always clear what the authors mean with “root tip area”; root cap? Tip area? Including elongation zone? How do they explain their findings and those of Ref. nr. 30, where it is explicitly stated that cutin/wax is found in the root cap but not elsewhere in the root?

Our response: We have modified the description in the revised manuscript to make it more precise. For instance, we used 5-mm-long root tips to extract RNA, cell wall, cuticle, or wax.

Using FY staining, we observed the deposition of cuticle not only in the root cap region but also in the elongation zone (Fig. 4). We carefully examined the depositions of cuticle and wax by TEM from root cap to elongation zone. We found some discontinuous cuticle in the elongation zone, but did not find any wax in the elongation zone. Our FY staining analysis also revealed the existence of cuticle in the elongation zone which is consistent with our TEM results. This is clarified in the revised manuscript.

8. While it is written in a generally understandable way, I think that the manuscript would benefit from a professional English editing to bring the language to the same level as the science presented.

Our response: Thank you very much for the suggestion. We have carefully revised the entire manuscript and the current version should be very much improved.

9. The discussion does not address the points that don't go well together but rather turn around Darwin and his studies with plants – which is historically interesting but does not provide a better insight into the findings described in the paper. They also seem very convinced that the three RLKs really directly have the described effects on all the processes (e.g. line 246). A less biased view and alternative explanations would be useful.

Our response: We revised the discussion part and tried our best to address the facts which are supported by our observations.

10. The title should be changed such that the effect of the RLKs on cutin/wax formation and FA metabolism is stressed. It appears that this is the primary effect – at least based on the data provided.

Our response: Thanks. We have changed the title in the revised manuscript. The new title is “Three receptor-like kinases negatively regulate root hydrotropism via directly controlling the biosynthesis of cell wall and its depositions in *Arabidopsis thaliana*”.

Minor points.

1. Lane 36-40 Auxin should not be involved in hydrotropic growth. But then, the *pin2* mutant (which changes auxin distribution) is affected in hydrotropic growth. That doesn't stick.

Our response: We and others found that auxin is not the primary determinant of root hydrotropism in *Arabidopsis*. Our previous experiment indicated *pin2* showed a higher root hydrotropic response because of the interference of gravitropism to hydrotropism. Removal of gravitropism can enhance hydrotropism. The roots of *pin2* mutant showed a greatly reduced root gravitropic response. This point has been clarified in the text (Page 3, line 37-42).

2. Lane 41-45. What is the activity MIZ1? That and the biological relevance to the topic is not clear.

Our response: *mizu-kussei 1 (miz1)* contains a recessive mutation in *At2g41660*. *miz1* roots show no hydrotropic response (Kobayashi et al., 2007). *miz1* plants exhibit no developmental or root gravitropic defects. Overexpression of *MIZ1* is able to enhance root hydrotropism (Miyazawa et al., 2012). The protein function of *MIZ1* is not known. What we know is that it contains a conserved domain with uncharacterized function (DUF617 domain). It is localized to the cytosolic side of the endoplasmic reticulum (ER) membrane (Yamazaki et al., 2012). *MIZ1* is not directly involved in the three LRR-RLK mediated hydrotropic response. But we have included the

information in the introduction part (Page 3, line 42-48). More information was added in the revised version of the manuscript.

3. There's a mix up of supplementary figures. In Fig. S2, a phylogenetic tree is shown that is not corrected mentioned in the text. In any case, I doubt that this is an extremely helpful figure given that it is not a point in the text.

Our response: Thanks. We corrected the issue in the revised manuscript (Page 7, line 112-116).

4. Lane 244. Expansins and Extensins are very different proteins with very different activities!!

Our response: We have carefully checked the entire manuscript and fixed the mistakes.

5. Suppl. Fig. 7: The section on cytokinins can be removed. It opens yet another avenue but is not being followed up in any way. As is, it is very insulated and does not provide much insight.

Our response: Thanks. We agree with you and removed that figure in the revised manuscript.

References

Kobayashi, A., A. Takahashi, Y. Kakimoto, Y. Miyazawa, N. Fujii, A. Higashitani, and H. Takahashi. 2007. A gene essential for hydrotropism in roots. *Proceedings of the National Academy of Sciences of the United States of America* 104:4724-4729.

Miyazawa, Y., T. Moriwaki, M. Uchida, A. Kobayashi, N. Fujii, and H. Takahashi. 2012. Overexpression of MIZU-KUSSEI1 enhances the root hydrotropic response by retaining cell viability under hydrostimulated conditions in *Arabidopsis thaliana*. *Plant & cell physiology* 53:1926-1933.

Yamazaki, T., Y. Miyazawa, A. Kobayashi, T. Moriwaki, N. Fujii, and H. Takahashi. 2012. MIZ1, an essential protein for root hydrotropism, is associated with the cytoplasmic face of the endoplasmic reticulum membrane in *Arabidopsis* root cells. *FEBS letters* 586:398-402.

Reviewer #3 (Remarks to the Author):

Summary

The manuscript by Chang et al. presents findings on the function of three RLKs in root hydrotropism through modulation of root tip cell walls and cuticle as well as asymmetric distribution of cytokinins during hydrotropism responses. Gain and loss of function lines were employed for phenotypic measures of root hydrotropism responses. A triple knockout, *arh1-2fei1-Cfei2-C*, that was generated through CRISPR-Cas9 editing of *FEI1* and *FEI2* in the *arh1-2* background was used for the majority of biochemical, microscopic, and molecular-genetic characterizations. The *arh1-2fei1-Cfei2-C* triple mutant was found to have hypersensitive hydrotropic responses whereas as individual overexpression lines exhibited a lack of or reduced hydrotropic responses. Further analyses showed a reduced deposition of cell wall and cuticle in the *arh1-2fei1-Cfei2-C* triple mutant as well as reduced transcript abundance of cell wall, lipid, and cuticle biosynthesis gene transcript abundance. Lastly, analysis of cell wall and cuticle-defective mutants and pectinase-treated WT plants revealed elevated hydrotropic responses and increased sensitivity to osmotic stress.

Review

The authors have presented an interesting collection of phenotypic, biochemical, and molecular-genetic characterization of these three RLKs, especially the *arh1-2fei1-Cfei2-C* triple mutant. The observed phenotypes, especially the biochemical phenotypes, are quite exceptional. I believe that the findings present herein have potential to improve our understanding of the relationship between cell wall modifications and root physiological responses. Furthermore, the results point to a potentially novel discovery on the regulation of cell wall through RLK-mediated signaling.

With that said, I do have some major concerns that put into question some of the conclusions that are presented in the manuscript.

1. First, the RNA-seq data merit further analyses and presentation of those analyses. It is important for the reader to be able to observe and assess all differentially expressed genes (DEGs) from the RNA-seq data set, as supplemental data, instead of only select genes of interest. This is particularly important given that the primary genes under investigation are Receptor Like Kinases that have the potential to regulate multiple biosynthetic and signaling pathways. I believe that this is required for the reader to be able to fully assess the conclusions drawn in the manuscript. It also isn't clear if the RNA-seq data have been deposited to NCBI's Sequence Read Archives (SRA) or a similar repository. This is generally a standard practice for manuscripts that present RNA-seq data.

Our response: We have added the whole RNA-seq data in the source data file. We actually did KEGG analysis and compared the expression profiles of wild type and the triple mutant. We found that the expression levels of the genes in the triple mutant involved in sugar metabolism and fatty acid biosynthesis are most significantly altered in the entire genome. Therefore, we did RT-qPCR for all the genes encoding for the enzymes catalyzing the biosynthesis of cell wall, pectin, cuticle, and wax. It has been clarified in the text (Page 13, line 240-247).

2. The presented lipidomics data are particularly concerning. No description of the lipid extraction methods is provided. This is particularly relevant for the presented classes and molecular species of lipids presented in the manuscript (Figure S18). Phosphatidylcholine (PC) should be the dominant class of lipids in roots (and other Arabidopsis tissues). In roots, phosphatidylcholine typically constitutes ~45 mole % of total root lipids (Li-Beisson et al., 2013, Acyl Lipid Metabolism, The Arabidopsis Book). No PC is presented in the lipidomics data (Figure S18). Furthermore, there appears to be a large preponderance of Phosphatidic Acids (PA), Lyso-lipids (lyso-phosphatidic acid, lyso-phosphatidylethanolamine, lyso-phosphatidylmethanol, lyso-phosphatidylethanol), phosphatidylmethanol and phosphatidylethanol. These are clear indications of improper extraction of the lipids. It is critical to inactivate the phospholipases by quenching the tissues with hot iso-propanol. In roots, there should

be close 0% PA (Li-Beisson et al., 2013, Acyl Lipid Metabolism, The Arabidopsis Book). Furthermore, the presence of lyso-phosphatidylmethanol is another clear indication of improper lipid extraction. "...phosphatidylmethanol which can be formed by the action of endogenous phospholipase D when plant tissue is extracted with methanolic solvents at or above room temperature (Roughan et al., 1978)" (Miquel & Browse, 1992, JBC, Vol. 267, No. 3, pp. 1502-1509). As such, no conclusions can be drawn from the lipidomics data.

Our response: The lipidomics analysis was repeated using an improved method suggested by the reviewer and the new results are presented in Supplementary Fig. 23. The methods used to extract the lipids from 5-mm-long root tips are now added in the material and methods (Page 27, line 518-530). A total of 491 lipids were identified in Col-0 and *arh1-2 fei1-C fei2-C*. Among all the lipids detected, 109 lipids are significantly altered in the triple mutant comparison with wild type ($P < 0.05$). The results are from six biological replicates (Supplementary Fig. 23).

Supplementary Fig. 23 | ARH1, FEI1, and FEI2 modulate lipid metabolism in root tips.

Relative amount of lipids in 5-mm root tips of wild-type (Col-0) and *arh1-2 fei1-C fei2-C*, which are presented by heatmaps according to the lipidomic analysis (n=6), $P < 0.05$. GL, glycerolipids; DG, diglyceride; TG, triglyceride; SL, saccharolipids; GP, glycerophospholipids; PC, phosphatidylcholine; LPC, lyso-phosphatidylcholine; MePC, methylphosphatidylcholine; PG, phosphatidylglycerol; PS, phosphatidylserine; PA, phosphatidic acid; PE, phosphatidylethanolamine; PET, phosphatidylethanol; SP, sphingolipids; Cer, ceramides. Data are the means of six biological replicates.

3. Lines 206 – 209: The authors state, “Surprisingly, we found a layer of electron-transparent cell wall modification covering the cuticle in the root caps of Col-0. We assumed that it is wax. This assumption was confirmed by using a *kcs7 kcs15 kcs20 kcs21* quadruple mutant as a negative control, in which the wax biosynthesis pathway is blocked and such a layer was no longer observed in the quadruple mutant (Fig. 3l, p, and t)”. The electron transparent layer that is being referred to (Figure 3m for WT) looks more like a fracture in the epoxy resin than a true anatomical feature of the plant. It is quite common for epoxy resin to fracture near hydrophobic surfaces. To support this conclusion, the authors would need to present multiple images from multiple, independent biological samples.

Our response: Thanks for the reasonable concern and good suggestion. We did TEM analyses at least ten times within the past two years. We are fully confident that the transparent layer we observed in wild type were not from epoxy resin fracture. First, we found this transparent layer mainly in wild type and in the cuticle mutant *gso1 gso2*, but not in the wax mutants (Fig. 3). In triple mutant *arh1-2 fei1-C fei2-C*, this layer is almost undetectable. Second, the root cap wax was disappeared after wax extraction (washed with chloroform for 30 seconds) in two-day-old Col-0 roots (Fig. 4c, d). The root cap wax is apparently synthesized in the cytoplasm of the outermost layer of root cap cells and form a drop-like structure, which subsequently transported and deposited to the outer layer of cuticle (Supplementary Fig. 20). Third, in the root tip wax extracts, we analyzed their compositions and they are indeed wax (Fig. 4a, b).

Fig. 4 | Amount of total and each chemical composition of wax and cuticle in the root tips of wild type, overexpressors, and various mutants.

a, Amount of total wax in different genotypes. **b**, Amount of each chemical composition of wax in different genotypes. **c**, **d**, TEM images showing cell wall, cuticle, and wax at the surface of the outermost lateral root cap cells (100 μm from the root tip) before and after wax extraction. **e**, Amount of total cutin in various genotypes. **f**, Amount of each chemical composition of the cutin in various genotypes. The measurements were carried out by using gas chromatography-mass spectrometer (GC-MS) (a, b, e, f). FA, fatty acids; HFA, ω -hydroxy fatty acids; TriOH 18:1 FA, 9,10,18-triOH C18:1 fatty acids; DCA, dicarboxylic acids. Data are the means \pm SD of three biological replicates (a, b, e, f). Scale bars represent 500 nm in (c, d). One-way ANOVA with Tukey's multiple comparison test was used for statistical analyses with $P < 0.01$.

Supplementary Fig. 20 | Wax is first accumulated in the cytoplasm of the outmost cells of the root cap and then transported to the out layer of cuticle.

a, The root cap wax is apparently accumulated in the cytoplasm and later transported to the outermost cell layer of the root caps, as revealed by transmission electron microscopy (TEM) analysis. The red arrow indicates the depositions of wax on the outer surface of the cuticle. Dot cycles indicate the wax drop in the cytoplasm of lateral root cap cells. **b**, The drop of wax was observed in the cytoplasm of the lateral root cap cells. **c-e**, The drop of wax is transported from lateral root cap cells to the outside layer of cuticle. Three biological replicates were carried out. Scale bars represent 5 μm in (a) and 500 nm in (b-e).

4. Furthermore, the authors may want to revise their understanding of the basic anatomy of plant cuticles. Plant cuticles are comprised of a cutin polymer that is embedded and covered with waxes. The cuticle is subdivided into different layers including the cuticular layer which represents the transition between a carbohydrate-rich cell wall and lipidic cutin polymer, the cuticle proper which is comprised largely of cutin that is embedded with intracuticular waxes, and an epicuticular wax layer (Yeats & Rose, 2013, *Plant Physiology*, 163, pp. 5–20). It is very difficult to ascertain what is “wax” from TEM images. Epicuticular waxes can be visualized with SEM.

Our response: Thanks. But from our TEM analysis, we can clearly distinguish cell wall, cuticle, and wax at the outmost layer of the root cap. The anatomy of the cell wall depositions could be slightly different between the root cap cells and the cells in the aerial parts of plants. As a matter of fact, the wax structure in the root cap cells was not reported in the literature based on our knowledge.

5. Line 342: “On one hand, thicker cell wall, cuticle, and wax...” Wax is part of the cuticle. The cuticle is comprised of a cutin polymer that is embedded and covered with cuticular waxes.

Our response: Thanks again. This question is similar to aforementioned other questions which we already answered.

6. To the best of my knowledge, a cuticular wax phenotype for the *kcs7 kcs15 kcs20 kcs21* quadruple mutant has never been verified (not in leaves, stems, or, relevant to this manuscript, root cap cuticles). The paper cited for this mutant refers to a *kcs7 kcs15 kcs21* triple mutant that has defects in pollen tapetal lipid deposition and has not been characterized for other cuticular phenotypes (i.e. root cap cuticle) (Zhang et al., 2021, *Front. Plant Sci.*, Sec. *Plant Physiology*, Volume 12, <https://doi.org/10.3389/fpls.2021.770311>). I think that it would behoove the authors to provide evidence and characterization of the cuticular defects of the *kcs7 kcs15 kcs20 kcs21* quadruple mutant.

Our response: This is a very good suggestion. In our revised version, the TEM images and new chemical analysis results of wax and cuticle are added which demonstrate that root caps of *kcs7 kcs15 kcs20 kcs21* quadruple mutant lack a layer of wax but are covered with a thicker layer of cuticle (Fig. 3p, t, Fig. 4, and Supplementary Fig. 19).

Fig. 4 | Amount of total and each chemical composition of wax and cuticle in the root tips of wild type, overexpressors, and various mutants.

a, Amount of total wax in different genotypes. **b**, Amount of each chemical composition of wax in different genotypes. **c**, **d**, TEM images showing cell wall, cuticle, and wax at the surface of the outermost lateral root cap cells (100 µm from the root tip) before and after wax extraction. **e**, Amount of total cutin in various genotypes. **f**, Amount of each chemical composition of the cutin in various genotypes. The measurements were carried out by using gas chromatography-mass spectrometer (GC-MS) (a, b, e, f). FA, fatty acids; HFA, ω-hydroxy fatty acids; TriOH 18:1 FA,

9,10,18-triOH C18:1 fatty acids; DCA, dicarboxylic acids. Data are the means \pm SD of three biological replicates (a, b, e, f). Scale bars represent 500 nm in (c, d). One-way ANOVA with Tukey's multiple comparison test was used for statistical analyses with $P < 0.01$.

Supplementary Fig. 19 | Compositions and amount of root cap wax.

a, Carbon-chain-length distributions of alkanes in wax extracted from the 5 mm root tips of Col-0 and other genotypes. **b**, Carbon-chain-length distributions of alcohols, ketones and fatty acids in wax extracted from the 5 mm root tips of Col-0 and other genotypes. Data are the means \pm SD of three biological replicates. One-way ANOVA with Tukey's multiple comparison test was used for statistical analyses with $P < 0.01$.

7. This also highlights another important consideration of the manuscript. The cuticle work is primarily focused on cuticular wax. However, the presented RT-qPCR illustrates downregulation of several cutin-related biosynthesis genes yet no chemical characterization of the root cap cutin of the *arh1-2fei1-Cfei2-C* triple mutant or *kcs7 kcs15 kcs20 kcs21* quadruple mutant is presented. Again, the authors have not clearly delineated the difference between cutin and wax which may be why these data are not presented. Given that cuticle is a major focus of the manuscript, it would perhaps be best to present some characterization of the cutin composition of the *arh1-2fei1-Cfei2-C* triple mutant and *kcs7 kcs15 kcs20 kcs21* quadruple mutants.

Our response: Thanks for the question. We analyzed the compositions and amount of cuticle and wax extracted from the root tips of Col-0, triple mutants, overexpressors of the three *LRR-RLKs*, *kcs7 kcs15 kcs20 kcs21* quadruple mutants, as well as *gso1 gso2* double mutants. Significant differences were observed in the chemical composition of root cap wax and cuticle. The root cap wax is predominantly composed of alkanes, alcohols, ketones, and fatty acids, whereas the cuticle was primarily composed of fatty acids and hydroxylated fatty acids. Interestingly, the cuticle related mutant *gso1 gso2* exhibited a significantly increased amount of wax. The mutant *kcs7 kcs15 kcs20 kcs21* contains more cuticle. (Fig. 4, and Supplementary Fig. 19). These results make perfect sense because both cuticle and wax are synthesized from the fatty acids as the precursors.

8. Lines 319 – 321: the authors refer to a potential function for ARH1-mediated signaling to control the expression of “yet to be identified” transcription factors. I encourage the authors to revisit the RNA-seq data and look for transcription factors known to regulate cuticle, cell wall, etc. Inclusion of a CSV or Excel file with the DEGs would be helpful.

Our response: We added the RNA-seq data into the source data file in the revised manuscript. We searched for the transcription factors known to regulate the biosynthesis of cell wall, cuticle, or wax but were unable to identify them among the

differentially expressed genes (DEGs). It is not surprising, as transcription factors downstream the RLK signaling pathways are often regulated through posttranslational modifications, such as phosphorylation.

9. Lines 332 – 337: The authors state, “Interestingly, we also identified a layer of electron-transparent cell wall modification laying outside of the cuticle which was only identified in the root cap region but not in the elongation zone (Fig. 3i-t). We then confirmed this identification by mutant analysis and composition determination by GC-MS that this layer of modification is actually wax.” This isn’t entirely accurate. Please see my comments about the TEM above. The transparent layer looks like a fracture in the epoxy resin and it is difficult to ascertain what is a “wax” layer from TEM. To verify this, many sections from multiple biological replicates would be required. If this transparent layer truly is “wax”, TEM analysis before and after wax extraction would be required to validate this conclusion.

Our response: Thanks for your questions. We did treat your questions seriously. We repeated at least ten times and similar results were obtained. We also compared the TEM images before and after wax extraction and found wax is removed after extracted by the chloroform (Fig. 4c, d). Our biochemical analysis also indicated that the transparent layer found in wild type and in *gso1 gso2* mutant is indeed the wax, not the fracture in the epoxy resin (Fig. 4).

Fig. 4 | Amount of total and each chemical composition of wax and cuticle in the root tips of wild type, overexpressors, and various mutants.

a, Amount of total wax in different genotypes. **b**, Amount of each chemical composition of wax in different genotypes. **c**, **d**, TEM images showing cell wall, cuticle, and wax at the surface of the outermost lateral root cap cells (100 μm from the root tip) before and after wax extraction. **e**, Amount of total cutin in various genotypes. **f**, Amount of each chemical composition of the cutin in various genotypes. The measurements were carried out by using gas chromatography-mass spectrometer (GC-MS) (a, b, e, f). FA, fatty acids; HFA, ω -hydroxy fatty acids; TriOH 18:1 FA, 9,10,18-triOH C18:1 fatty acids; DCA, dicarboxylic acids. Data are the means \pm SD of three biological replicates (a, b, e, f). Scale bars represent 500 nm in (c, d). One-way ANOVA with Tukey's multiple comparison test was used for statistical analyses with $P < 0.01$.

10. Subcellular localization experiments, Figure 2: Although it is expected that RLKs would localize to the PM, the presented confocal images aren't sufficient to demonstrate PM localization. A PM marker should be used to validate co-localization.

Our response: In our revised version, the subcellular localization of ARHs was observed by plasmolysis using transgenic plants harboring *35S::ARH1-YFP*, *35S::FEI1-YFP*, or *35S::FEI2-YFP*. The results showed all these RLKs were plasma membrane localized (Supplementary Fig. 2b).

Supplementary Fig. 2 | ARH1, FEI1, and FEI2 are localized on the plasma membrane.

a, A phylogenetic tree of ARH1, FEI1, and FEI2 based on their full-length amino acid sequences. BRI1 was used as an outgroup control. **b**, The subcellular localizations of ARH1-YFP, FEI1-YFP, and FEI2-YFP were determined in the root elongation zones of transgenic plants overexpressing their corresponding coding genes. Plasma membrane localization was determined by plasmolysis using 800 mM D-sorbitol. Three biological replicates were carried out. Scale bar represents 10 μm .

11. Subcellular localization experiments, Figure S13, Lines 178-179: “The distribution and localization of ARH1-YFP, FEI1-YFP, and FEI2-YFP were not altered by hydrostimulation treatment (Supplementary Fig. 13).” It is very difficult to ascertain this from the presented images. Based on the images presented in Figure S13, hydrostimulation of pFEI2::gFEI2-YFP plants appears to stimulate promoter activity in cortical cells and maybe even vasculature. Clearer images would be helpful.

Our response: Thanks for the questions. New images are presented in the revised manuscript (Supplementary Fig. 15).

Supplementary Fig. 15 | The distributions and polar localizations of ARH1-YFP, FEI1-YFP, and FEI2-YFP were not altered after hydrostimulation treatment.

a-f, The distributions and polar localizations of three LRR-RLKs before and after hydrostimulation treatment. Four-day-old transgenic seedlings harboring

pARH1::gARH1-YFP, *pFEI1::gFEI1-YFP*, or *pFEI2::gFEI2-YFP* were treated with control (a, c, e) or hydrostimulation with 200 mM D-sorbitol at the bottom right side of the medium (b, d, f). **g-i**, YFP fluorescence ratio between the right and left sides (controls), or between convex and concave sides within a 400- μ m root tip starting from the quiescent center (hydrostimulated seedlings) was analyzed (as depicted in figure c). **j**, A representative root tip of *pARH1::gARH1-YFP* after 1 hour hydrostimulation treatment, red arrows indicate the polar localization of *ARH1-YFP* facing to the newly formed cell plate. **k**, Differences of the number of newly formed cortex cell plates of right side versus left side (controls), or between convex and concave sides in the meristematic zone (hydrostimulated seedlings). Scale bars represent 50 μ m. “n” represents the number of roots analyzed in the experiment. Three biological replicates were carried out. Student’s *t*-test was used for statistical analyses with $P < 0.05$.

12. Methods, Fluorol yellow staining: What emission wavelength or filter was used for confocal imaging?

Our response: The emission wavelength used in the imaging of Fluorol yellow staining is 488 nm. The information is added in the revised manuscript (Page 25, line 480).

Minor concerns

1. Figure 1n, Lines 145 – 147. “Root tips of *arh1-2 fei1-C fei2-C* triple mutant were swelled after seedlings were transferred from normal 1/2 MS medium to a split 1/2 MS medium supplemented with 800 mM D-sorbitol....” The manuscript text says 800 mM, but the figure is labeled as 400 mM. Please clarify. Figure 1n is also lacking the red arrows that delimit the root zone used for cell counts.

Our response: We have carefully read through the entire manuscript and fixed the mistakes we made in the previous version (Page 9, line 154-157).

2. Line 268: KCS enzymes are referred to as “wax synthetases”. KCS enzymes are beta-ketoacyl CoA synthases.

Our response: Thanks. We have modified this statement.

Conclusions

Chang et al. have presented an interesting collection of phenotypes that potentially points to a role for ARH1 in regulating root hydrotropism through modulating cell wall and cuticle. However, there are some major concerns that should be addressed to provide better support for the conclusions stated in the manuscript. I recommend the following to the authors: 1) to provide a more comprehensive analysis of the RNA-seq data including a CSV or Excel file with all DEGs, 2) to clearly differentiate between wax, cutin, and cuticle, 3) to employ a series of mutants defective specifically and uniquely in cutin, wax, specific cell wall carbohydrate domains, membrane lipids, etc. to better understand the relative contribution of each to root hydrotropic responses, 4) to analyze the cutin content of arh1-2 fei1-C fei2-C triple mutant, 5) to characterize the kcs quadruple mutant in terms of wax and cutin content and composition (and employ mutants specifically affected in cutin versus wax), 6) to provide better support for PM localization of ARH1, FEI1, and FEI2 through the use of PM markers, and 7) repeat the lipidomics with proper extraction conditions (or utilize simpler lipid analyses with proper extraction conditions). Further analysis of the RNA-seq data will be revelatory and may present opportunities for a better and more comprehensive investigation of the function of ARH1, FEI1, and FEI2 through further experimentation. This may alter the trajectory of the work and will likely require further discussion.

Our response: Thanks. We carried out all the experiments you suggested and changed the context accordingly. Cuticle is an electron-opaque layer covering root tips. The chemical composition of this layer is referred to as cutin. For suggestion 3,

we investigated the root hydrotropism of mutants *cesa1*, a mutant of a cellulose synthase gene, and *bdg*, which was previously reported as a cuticle defective mutant. Roots of *cesa1* and *bdg* exhibited enhanced hydrotropic responses, consistent with our findings described in this manuscript (Supplementary Fig. 24).

Supplementary Fig. 24 | Mutants with defects in cell wall or cuticle show enhanced response to moisture gradient.

a, b, Hydrotropic responses of wide-type (Col-0) and mutants with defects in cell wall (cellulose) and cuticle. **c**, Root growth curvatures of *cesa1* and *bdg* after 24-hour hydrostimulation treatments. Each circle represents the measurement of an individual root. “n” represents the number of roots analyzed in the experiment. Three biological replicates were carried out. One-way ANOVA with Tukey’s multiple comparison test was used for statistical analyses with $P < 0.01$.

Reviewer #1 (Remarks to the Author):

Comments to the authors

This is the second review of this manuscript, and I felt some of my concerns were properly answered. I appreciate the authors for adding new data for my comment. However, there remains a few problems that should be clearly corrected.

Major concern

1. In my first review, I suggested the authors to use the term "regulate" more carefully. However, the amendments do not seem to be completed. First of all, the title is not still appropriate. One can still read as three RLKs negatively regulate root hydrotropism. I think the title might be, for example, "Defects in wax and cutin production mediated by the three RLKs alter root hydrotropism in *Arabidopsis thaliana*". I do not think that the title I wrote is as the best one, however the current title is still misleading.
2. I cannot understand the reason why the authors used only 200mM sorbitol in newly added figure S7. In Fig1i-n, they demonstrated that viability of root cells was less affected by 200mM sorbitol. Additionally, hydrotropic response seemed to be enhanced as the concentration of sorbitol increased (Fig. 1d and h). Observing the effect of higher sorbitol concentration will strongly support the authors' idea as well as help the readers evaluating the effect of gravitropism on hydrotropism in the authors' experimental system.

Minor concerns

1. Concerning to the newly added sentences (lines 39-42), I think the reference is not adequate. If the authors wish to mention the second sentence (lines 41-42), papers that comprehensively describes the role of auxin in hydrotropism should be added. In Takahashi et al. 2012 (Ref 23), they used several auxin-related mutants including an allele of *pin2*, and found that root hydrotropic response is observed in the mutants. Alternatively, Kaneyasu et al. 2007 (Ref. 9) Shkolnik et al. 2016 (Ref. 11) are also relevant to the description. These sentences should be rewritten.
2. Answer to the minor comment 1: Although the reference is restricted to the report on *Arabidopsis*, the fact should be clearly mentioned in the main text; e.g. A number of studies using *Arabidopsis* indicated that ..., Takahashi's group identified two nucleotide-substitution mutants of *Arabidopsis*,
3. Some sentences remain illegible. Please carefully check and make the manuscript grammatical.

Reviewer #2 (Remarks to the Author):

The manuscript by Chang et al is a revised version describing the role of three LRR-RLKs of *Arabidopsis* in hydrotropic growth. Technically, the authors have responded to my major concerns by showing additional experiments. There are still a number of major issues that require significant improvement.

They should define at first mention of "...the cell wall and its depositions...", depositions refers to cuticle and wax. At this point, they are repeating this over and over again, which is not necessary. In the introduction, they mention cytokinin, which is not necessary, since they have removed these experiments from the manuscript. They also mention *miz1/miz2/gnom* without giving any context. They should better introduce the LRR-RLKs. How many are encoded in *Arabidopsis*, are ARH1, FEI1 and FEI2 of the same phylogenetic group, i.e. very similar or not? They also show data with SOS5, but fail to mention what this is and where/how SOS5 connects to FEI1 and FEI2. What is the relevance of ARF7 and suomylation in the context of their results? Why mention the compensatory effects of *cesa* mutants (lane 73-77)?

Lane 249 and figure legend to panel A; what is "rich ration", what is Q value?

Lane 258: the FEI2 downstream genes are not explained. This comes out of the blue. How were these identified, characterized? There is a lack of context.

Lane 281: what are GSO1 and GSO2? Are they related to the other LRR-RLKs? And what is BGD?

Lane 97 and elsewhere, e.g. 322: I insist that the authors do not provide evidence that the LRR-

RLKs are involved in the biosynthesis of cell wall components (for sure not pectin, which is produced in the Golgi). They might be involved in the regulation of cell wall biosynthesis. In the suppl Fig 9, there is a quite obvious increase in root diameter, even under control conditions. The authors claim to see no difference – which is not what I believe to see in the panels.

Minor:

Lane 112: In total, we identified..., also lane 240

Lane 142: reformulate ...they are not edited mutants.

Lane 157: reformulate.... Osmotic stress medium applied... Same for the following sentence. Same for lane 185-188.

Lane 236: the composition cannot be decreased. Reformulate.

Lane 303: "the treatment of osmotic stress". I guess osmotic stress is the treatment. Reformulate

Lane 310 and elsewhere: facts should be written in present tense.

And many more small mistakes that need to be corrected.

Reviewer #3 (Remarks to the Author):

Dear Authors,

I am impressed with the amount of work that was done to address my, and the other reviewer's, concerns. I am largely satisfied with the manuscript with the sole exception that the authors are using the word cuticle to refer to cutin, the polymeric component of the cuticle. Cuticle = cutin + wax. I recommend correcting this.

I believe that the authors have made an important discovery.

Kind regards,
Reviewer #3

Responses to reviewers

Dear Reviewers

Thank you for your comments concerning our manuscript previously entitled “Three receptor-like kinases negatively regulate root hydrotropism via directly controlling the biosynthesis of cell wall and its depositions in *Arabidopsis thaliana*” (NCOMMS-23-23839A). All comments are invaluable for us to further revise and improve our manuscript. We have treated these comments very carefully and amended our manuscript accordingly. Here list our corrections in details.

Reviewer #1 (Remarks to the Author):

Comments to the authors

This is the second review of this manuscript, and I felt some of my concerns were properly answered. I appreciate the authors for adding new data for my comment. However, there remains a few problems that should be clearly corrected.

Major concern

1. In my first review, I suggested the authors to use the term “regulate” more carefully. However, the amendments do not seem to be completed. First of all, the title is not still appropriate. One can still read as three RLKs negatively regulate root hydrotropism. I think the title might be, for example, “Defects in wax and cutin production mediated by the three RLKs alter root hydrotropism in *Arabidopsis thaliana*”. I do not think that the title I wrote is as the best one, however the current title is still misleading.

Our response: Thank you very much for your suggestion. We now changed our title to “Defects of the cell wall and its depositions caused by loss-of-function of three RLKs alter root hydrotropism in *Arabidopsis thaliana*”.

2. I cannot understand the reason why the authors used only 200 mM sorbitol in newly added figure S7. In Fig1i-n, they demonstrated that viability of root cells was less affected by 200mM sorbitol. Additionally, hydrotropic response seemed to be enhanced as the concentration of sorbitol increased (Fig. 1d and h). Observing the effect of higher sorbitol concentration will strongly support the authors' idea as well as help the readers evaluating the effect of gravitropism on hydrotropism in the authors' experimental system.

Our response: Sorry for the confusion. Our hydrostimulation experiments were performed on a split-agar medium with the right bottom side of the medium containing different concentrations of D-sorbitol. The root tips were initially placed on the regular 1/2 MS medium 0.5 cm away from the border. The osmotic stress experiments were performed directly on the medium containing D-sorbitol. The detailed experimental approaches were described in Materials and methods (line 428 and line 455). For these reasons, root tips on the hydrostimulation medium can tolerate higher concentration of D-sorbitol, but cannot tolerate the D-sorbitol concentration higher than 400 mM (Supplementary Fig. 9). Our experimental results indicated that the root tips cannot grow under a 400 mM D-sorbitol osmotic stress treatment. Therefore, higher concentrations of D-sorbitol were not utilized in the experiments presented in Supplementary Fig. 7.

Supplementary Fig. 9 | Root tip cells of the triple mutants exhibited abnormal expansion upon hydrostimulation or osmotic stress treatment.

a-h, Phenotypes in the root tips of Col-0 after treated with hydrostimulation or osmotic stress. Four-day-old Col-0 seedlings were transferred from 1/2 MS medium to split 1/2 MS media, supplemented with various concentrations of D-sorbitol at the bottom right side of the medium (a-d), or to osmotic stress media containing different concentrations of D-sorbitol (e-h) and incubated for 24 hours. **i-p**, Phenotypes in the root tips of the triple mutant after treated with hydrostimulation or osmotic stress. Four-day-old *arh1-2 fei1-C fei2-C* seedlings were transferred from 1/2 MS medium to split 1/2 MS media, supplemented with different concentrations of D-sorbitol at the bottom right side of the media (i-l) or to osmotic stress media containing different

concentrations of D-sorbitol (m-p) and incubated for 24 hours. Three biological replicates were carried out. Scale bars represent 50 μm .

Minor concerns

1. Concerning to the newly added sentences (lines 39-42), I think the reference is not adequate. If the authors wish to mention the second sentence (lines 41-42), papers that comprehensively describes the role of auxin in hydrotropism should be added. In Takahashi et al. 2012 (Ref 23), they used several auxin-related mutants including an allele of *pin2*, and found that root hydrotropic response is observed in the mutants. Alternatively, Kaneyasu et al. 2007 (Ref. 9) Shkolnik et al. 2016 (Ref. 11) are also relevant to the description. These sentences should be rewritten.

Our response: Thank you very much for your suggestions. We have revised these sentences, ensuring that the references are accurately cited (lines 39-43).

2. Answer to the minor comment 1: Although the reference is restricted to the report on Arabidopsis, the fact should be clearly mentioned in the main text; e.g. A number of studies using Arabidopsis indicated that ..., Takahashi's group identified two nucleotide-substitution mutants of Arabidopsis,

Our response: Thanks. We have revised these sentences based on your suggestions (lines 36-39, lines 43-45).

3. Some sentences remain illegible. Please carefully check and make the manuscript grammatical.

Our response: Thank you very much for the suggestion. We have carefully revised the entire manuscript and the current version should be greatly improved.

Reviewer #2 (Remarks to the Author):

The manuscript by Chang et al is a revised version describing the role of three LRR-RLKs of Arabidopsis in hydrotropic growth. Technically, the authors have responded to my major concerns by showing additional experiments. There are still a number of major issues that require significant improvement.

1. They should define at first mention of "...the cell wall and its depositions...", depositions refers to cuticle and wax. At this point, they are repeating this over and over again, which is not necessary.

Our response: Thanks for your suggestion. We now defined the cell wall, cutin, and wax as CCW in the text (lines 18-20). In this way, we do not need to repeat cell wall, cutin and wax again and over again.

2. In the introduction, they mention cytokinin, which is not necessary, since they have removed these experiments from the manuscript. They also mention *miz1/miz2/gnom* without giving any context. They should better introduce the LRR-RLKs. How many are encoded in Arabidopsis, are ARH1, FEI1 and FEI2 of the same phylogenetic group, i.e. very similar or not? They also show data with SOS5, but fail to mention what this is and where/how SOS5 connects to FEI1 and FEI2. What is the relevance of ARF7 and suomylation in the context of their results? Why mention the compensatory effects of *cesa* mutants (lane 73-77)?

Our response: Thanks for your questions. The aforementioned issues have now been clarified in this revised manuscript. In the introduction, we presented a comprehensive overview concerning MIZ1 and RLKs (lines 46-50, lines 76-83). We included the information of SOS5 because its mutant shows a swollen root tip phenotype under high concentration of sucrose similar to the *fei1 fei2* double mutant. SOS5 is a fasciclin-like GPI-anchored extracellular arabinogalactan protein. The *fei1 fei2 sos5*

triple mutant did not exhibit additive root tip defects on 1/2 MS media supplemented with 4.5% sucrose, suggesting that FEI1, FEI2, and SOS5 regulate the response of the root tip to high concentrations of sucrose, likely through the same signaling pathway (lines 331-337). The discussion on the compensatory effects of *cesa* mutants has been revised (lines 74-75). In the introduction, we discussed the major discoveries regarding the mechanisms regulating root hydrotropism, including cytokinins and ARF7, even though they were not directly relevant to the research presented in this manuscript.

3. Lane 249 and figure legend to panel A; what is “rich ration”, what is Q value?

Our response: Thanks for your questions. We refined the calculation methodology and re-performed the KEGG analysis. The result is nearly identical (Fig. 5a). The “Rich Ratio” represents the degree of enrichment of DEGs in each pathway. The size of each circle represents the number of the enriched DEGs, with a larger circle indicating a greater number. The Q value represents the adjusted *P*-value after multiple testing, and a smaller Q value indicates a more significant enrichment effect. These issues are clarified in the figure legend of Fig. 5 (lines 869-873).

4. Lane 258: the FEI2 downstream genes are not explained. This comes out of the blue. How were these identified, characterized? There is a lack of context.

Our response: Thanks. It our mistake. We have revised them as FEI2 regulated genes. Their transcriptional levels were significantly down-regulated in *fei2-C* and *fei2* mutants, as demonstrated in Supplementary Fig. 21a, b (lines 270-273).

5. Lane 281: what are GSO1 and GSO2? Are they related to the other LRR-RLKs? And what is BGD?

Our response: GSO1 and GSO2 are two closely related LRR-RLKs. A mutant of *gso1 gso2* exhibited defects in cutin biosynthesis in the root tips and aerial parts of Arabidopsis. In a previous report, it was demonstrated that SERKs, a group of LRR-RLKs, function as co-receptors of GSO1 and GSO2 (Reference 47). The GSO1/GSO2-SERK complex is capable of perceiving the extracellular peptide TWS1 and regulate the integrity of embryonic cutin. BDG is an α/β -hydrolase family protein that localizes to the outermost portion of the epidermal cell wall (lines 297-298). These have been clarified in the text.

6. Lane 97 and elsewhere, e.g. 322: I insist that the authors do not provide evidence that the LRR-RLKs are involved in the biosynthesis of cell wall components (for sure not pectin, which is produced in the Golgi). They might be involved in the regulation of cell wall biosynthesis.

Our response: Thanks for your suggestions. We have modified these statements as “FEI1 and FEI2 are involved in the regulation of CCW biosynthesis” (lines 95-98, lines 229-231, lines 267-268, lines 339-342, and lines 352-354).

7. In the suppl Fig 9, there is a quite obvious increase in root diameter, even under control conditions. The authors claim to see no difference – which is not what I believe to see in the panels.

Our response: Thanks for your suggestions. We have revised this description (lines 160-162). The root diameter of the triple mutant is increased compared with that of Col-0. This phenotype can be observed in figures, including Fig.1 i-n, Supplementary Fig. 9, Supplementary Fig. 10, Supplementary Fig. 11, and Supplementary Fig. 14, and also in Movie 1 and Movie 2.

Minor:

Lane 112: In total, we identified..., also lane 240

Lane 142: reformulate ...they are not edited mutants.

Lane 157: reformulate.... Osmotic stress medium applied... Same for the following sentence. Same for lane 185-188.

Lane 236: the composition cannot be decreased. Reformulate.

Lane 303: "the treatment of osmotic stress". I guess osmotic stress is the treatment.

Reformulate

Lane 310 and elsewhere: facts should be written in present tense.

And many more small mistakes that need to be corrected.

Our response: Thank you very much for your suggestions. We have carefully revised the entire manuscript and the current version should be greatly improved.

Reviewer #3 (Remarks to the Author):

Dear Authors,

I am impressed with the amount of work that was done to address my, and the other reviewer's, concerns. I am largely satisfied with the manuscript with the sole exception that the authors are using the word cuticle to refer to cutin, the polymeric component of the cuticle. Cuticle = cutin + wax. I recommend correcting this.

I believe that the authors have made an important discovery.

Kind regards,

Reviewer #3

Our response: I would like to express my sincere gratitude for your invaluable advice and recognition. We fully endorse the concept that cuticle = cutin + wax, and have integrated this concept into this revised manuscript.

Reviewer #1 (Remarks to the Author):

I read the revised manuscript carefully and found that the authors had responded to my concerns. I appreciate the authors' effort.

Minor comment:

Authors should correct the author list of Ref. 5, for it seems that the full names of the authors are written.

Reviewer #2 (Remarks to the Author):

The manuscript by Chang et al is an extensive analysis of LRR-RLKs that play a role in root hydrotropism in Arabidopsis. The authors have addressed the points that I have raised in my previous comments. Apart from smaller issues, the manuscript has been greatly improved.

Minor points that I found:

Lane 41: ...mutants, show enhanced... (since this statement is an established fact, it should be written in present tense)

Lane 52: why can MIZ2 not regulate root hydrotropism? A few lanes above, the authors introduce the miz2 mutant as having been identified in a screen for aberrant moisture response. There is some information missing here. Alternatively, they could just say that the role of MIZ2 is not yet known. Unless it is known, but this point is not clarified in the text.

Lane 98: I would drop "strongly", I don't see how a transmembrane protein can strongly localize to the plasma membrane. Perhaps the authors mean to say that these RLK predominantly localize to the plasma membrane?

Lane 104: CCW is not the abbreviation of root cap wax. I assume that the term CCW needs to be introduced again in the introduction section, not only in the abstract.

Lane 108: We demonstrate..... not demonstrated

Lane 862: ... genes encoding the enzymes.... ORgenes coding for the enzymes....

Lane 868:... to cutin and wax are marked....

Lane 869:...."rich ratio" represents....

Lane 871:.... is represented....

Responses to reviewers

Reviewer #1 (Remarks to the Author):

I read the revised manuscript carefully and found that the authors had responded to my concerns. I appreciate the authors' effort.

Minor comment:

Authors should correct the author list of Ref. 5, for it seems that the full names of the authors are written.

Our response: Thank you very much. We have revised the manuscript and the error has been corrected.

Reviewer #2 (Remarks to the Author):

The manuscript by Chang et al is an extensive analysis of LRR-RLKs that play a role in root hydrotropism in Arabidopsis. The authors have addressed the points that I have raised in my previous comments. Apart from smaller issues, the manuscript has been greatly improved.

Minor points that I found:

Lane 41: "...mutants, show enhanced..." (since this statement is an established fact, it should be written in present tense)

Our response: Thanks. We have corrected the error.

Lane 52: why can MIZ2 not regulate root hydrotropism? A few lanes above, the authors introduce the miz2 mutant as having been identified in a screen for aberrant moisture response. There is some information missing here. Alternatively, they could just say that the role of MIZ2 is not yet known. Unless it is known, but this point is not clarified in the text.

Our response: Thanks for your suggestion. We have modified these statements as “Knocking out GNOM leads to a lethal phenotype, whereas knocking out MIZ1 does not cause any obvious developmental defects. These observations suggest that MIZ1 can specifically regulate root hydrotropism, and MIZ2 is involved in multiple biological processes.”

Lane 98: I would drop “strongly”, I don’t see how a transmembrane protein can strongly localize to the plasma membrane. Perhaps the authors mean to say that these RLK predominantly localize to the plasma membrane?

Our response: Thanks for your suggestion. We have deleted the word “strongly”.

Lane 104: CCW is not the abbreviation of root cap wax. I assume that the term CCW needs to be introduced again in the introduction section, not only in the abstract.

Our response: Thanks for your suggestion. We added the definition of CCW again in the introduction section.

Lane 108: We demonstrate... not demonstrated

Lane 862: ... genes encoding the enzymes... OR ...genes coding for the enzymes...

Lane 868:... to cutin and wax are marked...

Lane 869:... "rich ratio" represents...

Lane 871:... is represented...

Our response: Thank you very much for your suggestions. We have carefully revised the entire manuscript and these errors have been corrected.